# On the Cause of Large Daily River Flow Fluctuations in the Mekong River

Khosro Morovati[1,2], Keer Zhang[1,2], Lidi Shi[3], Yadu Pokhrel[4], Maozhu Wu[1], Paradis Someth[5], Sarann Ly[6], Fuqiang Tian[*1,2]

[1]Department of Hydraulic Engineering & State Key Laboratory of Hydro-science and Engineering, Tsinghua University, Beijing, 100084, China
[2]Key Laboratory of Hydrosphere Sciences of the Ministry of Water Resources, Tsinghua University, Beijing, 100084, China
[3]Department of Physical and Environmental Sciences, University of Toronto Scarborough, Toronto, Ontario, Canada
[4]Department of Civil and Environmental Engineering, Michigan State University, East Lansing, 48823, United States
[5]eWater, UC Innovation Centre, University Drive South, Canberra, Australian Capital Territory, 2617, Australia
[6]Mekong River Commission Secretariat (MRCS), Technical Support Division, Vientiane 01000, Lao PDR

*Correspondence to*: Fuqiang Tian (tianfq@mail.tsinghua.edu.cn)

**Abstract.** Natural fluctuations in river flow are central to the ecosystem productivity of basins, yet significant alterations in daily flows pose threats to the integrity of the hydrological, ecological, and agricultural systems. In the dammed Lancang-Mekong River (hereafter LMR), the attribution of these large daily flow changes to upstream regions remains mechanistically unexamined, a factor blamed on challenges in estimating the time required for large daily shifts in upstream river flow to impact the downstream stations. Here, we address this with a newly developed sub-basin modeling framework that integrates 3D hydrodynamic and response time models, as well as a hydrological model with an embedded reservoir module. This integration allows us to estimate the time required between two hydrological stations and to distinguish the contribution of sub-basins and upstream regions to large daily river flow alterations. Findings revealed a power correlation between upstream river discharge and the required time to reach downstream stations. Significant fluctuations (greater than 1 m) in the river's daily flow were evident before the advent of the era of human activities, i.e., before 1992, with around 92% of these fluctuations occurring during the wet season, particularly in June, July, and August. This pattern persisted throughout subsequent periods, including the growth period (1992-2009) and the mega-dam period (2010 to 2020), with minimal variation in the frequency of events. The Lancang basin contributed to approximately 33-42% of these large river fluctuations at the Chiang Saen station. We found that daily-scale water level and runoff might not fully capture dynamic river flow changes, as significant differences were observed between daily and sub-daily river flow profiles. Sub-basins significantly contributed to mainstream discharge, leading to substantial shifts in mainstream daily river flows. The outcomes and model derived from the sub-basin approach held significant potential for managing river fluctuations and have broader applicability beyond the specific basin studied.

# 1 Introduction

Natural flow regimes provide temporal and spatial fluctuations in river water level/flow, which are central to supporting productive environmental and ecological systems (Van Binh et al., 2020). However, large changes in river flow - mainly due to human intervention and climate change - pose a threat to ecosystem productivity and sustainable development, disrupting the integrity of rivers, causing bank erosion (Darby et al., 2013), leading to successive saturation and draining, and altering natural hydrological rhythms (Yoshida et al., 2020; Soukhaphon et al., 2021).

Two primary drivers of these hydrological alterations in the LMR basin are human activities (e.g., uncoordinated dam operations) and significant spatiotemporal variability in precipitation (Zhang et al., 2023; Yun et al., 2020; Wang et al., 2021a). Dam water storage and operations can alter peak flows, increase base flows, and modify the frequency and variability range of discharge (Hecht et al., 2019). For instance, since 2010, the Chiang Saen station in Thailand, near the China border, has recorded a 98% increase in monthly discharge during the dry season, while the wet season water level dropped by 1.55 meters (Lu and Chua, 2021). Intense downpours lasting several hours or days further exacerbate downstream flow alterations (Wang et al., 2017a). Undammed regions can deliver large discharges into the downstream areas, compounding the impact of these stressors and causing daily water level fluctuations of 1-4 meters (MRC, 2011). These fluctuations can influence critical phenomena like the flood pulse, which drives productivity in downstream regions such as the Tonle Sap Lake (Morovati et al., 2021a; Morovati et al., 2023), affecting agriculture and fishery (Sabo et al., 2017; Chen et al., 2021; Wang et al., 2021a). Additionally, these changes trigger fish mortality by confining fish to small water bodies, altering spawning patterns and fish migration, and affecting agricultural and livestock production (Burbano et al., 2020; Li et al., 2022; Morovati et al., 2024) - a concern recently raised by local communities. Despite the importance of these issues, research on assessing such large daily river flow fluctuations (1-4 m) remains limited.

Flow regime analysis has driven extensive research in recent years. Han et al. (2019) used CREST-snow hydrological model with remote sensing data and found a 6% change in mean annual streamflow at the Lancang River from 2008 to 2014. Wang et al. (2021b) applied SWAT hydrological model to project daily runoff until 2050, revealing increased flood risks in the lower Mekong. Shin et al. (2020) utilized the CaMa-Flood hydrodynamic model, accounting for 86 dams, and found that surface water storage was mainly governed by climate variation before 2010. Galelli et al. (2022) employed VIC and VIC-Res models to simulate dam re-operations, demonstrating a 39% increase in median daily flow in January and a 10% decrease in August in the lower Mekong. Yun et al. (2020), using the VIC-Res model, reported a 5% reduction in annual streamflow at Chiang Saen due to Lancang cascade dams from 2008 to 2016. Wang et al. (2021a) employed VIC and CaMa-Flood models to analyze daily floods in the LMR basin from 1976 to 2015. Most recently, Yun et al. (2024) used this model to investigate reservoir impacts on natural runoff, finding that by 2023, reservoirs could store up to 62% of the annual runoff under extreme conditions.

There are many other studies resulting in successful analyses of flow regime changes in the LMR basin, including historical assessment using indicators of hydrologic alteration (Cochrane et al., 2014, Lu et al., 2014; Li et al., 2017; Lu and Chua,

2021), and monthly assessments of Chinese dams' impacts on the downstream flow regime using VMod, a distributed

hydrological model (Räsänen et al., 2017). Additionally, studies have examined the impacts of constructed tributary dams in the lower Mekong (Piman et al., 2016) using SWAT and HEC-ResSim models. Furthermore, the rainfall-runoff-inundation model (RRI) has been applied to address climate change and reservoir operations' impacts on inundation patterns based on daily and monthly simulations of river runoff across the lower Mekong (Try et al., 2018, 2020, and 2022; Ly et al., 2023). However, none of these models and other developed tools for assessing hydrodynamics, hydrology, and sediment dynamics

in the LMR basin discussed in the review by Johnston and Kummu (2012), provide an assessment of the degree to which the downstream large daily water level/flow changes are attributed to upstream sub-basins and the required time for shifts in upstream river flow to impact downstream stations.

Here, we first identify the large daily river flow changes by analyzing observed historical data over the last four decades. We then address the gaps mentioned above by developing an integrated modeling framework consisting of a highly accurate 3D

hydrodynamic model (Delft3D-Flow) to simulate daily water level/flow and velocity, a response time model to explicitly attribute the daily river flow changes at mainstream stations to their respective sub-basin and upstream station(s), and a hydrological model to provide daily discharge for tributaries lacking measured data. Our analysis based on the developed models expands on previous research on at least three aspects: (i) our approach allows us to quantitatively assess the regional contribution to downstream abnormal water level/flow shifts. Indeed, this analysis shifts the current conversations from how

much water level/flow has historically altered - including small river flow changes existing even in undammed river basins - to how much upstream sub-basins have contributed to large daily water level/flow changes, which is significant for regional and transboundary development; (ii) the results offer essential insights into the time required for upstream river flow changes to propagate to the downstream station. This facilitates improved management strategies for sub-basins, crucial for mitigating abnormal flow regime changes that pose threats to communities residing near the mainstream; (iii) these models

and analyses provide insights into the concerns raised by locals regarding the roles of climate change and human activities in the large daily river flow fluctuations in the LMR. Furthermore, the findings and the developed model can serve as a reference for understanding similar issues in other basins.

## 2 Material and Methods

### 2.1 Study area

With ~4800 km in length, the pan-shaped LMR constitutes the second most diverse aquatic ecosystem globally (MRC, 2011; Intralawan et al., 2019) and ranks as the eighth largest in terms of annual runoff (Sabo et al., 2017). Its extensive length encompasses diverse geographical regions including deep valleys and lowland areas, which has facilitated both dam construction and agricultural development (Yoshida et al., 2020; Tian et al., 2023). The LMR is divided into two reaches, the upper course is known as the Lancang River within China where it originates from the Tibetan Plateau and is home to 11

large mainstream hydropower dams and many tributary dams (see Figure 1), and the lower reach is known as the lower

Mekong, where its surrounding sub-basins have been heavily impacted by agricultural activities and tributary dams (Zhang et al., 2023). Until the end of 2020, two mainstream hydropower dams operated in the lower Mekong (e.g., Don Sahong and Xayaburi); however, their total storage capacity is much smaller than some tributary dams, including Nam Ngum (4,700 MCM) and Xe Kaman (4,800 MCM) (see Figure 1). Although each tributary reservoir typically has a small storage capacity

(e.g., < 5 km³, see Figure S1 in the supplementary Material (SM)), their cumulative storage and independent operation, due to varying priorities among riparian countries, could intensify hydrological changes (Zhang et al., 2023). 10% of the total hydropower potential across the LMR basin is attributed to these mainstream hydropower reservoirs (MRC, 2019, Morovati et al., 2023).

The hydrology of the basin is mainly influenced by an uneven distribution of precipitation, both spatially and temporally

(Pokhrel et al., 2018). The wet season, occurring from June to November (LMC and MRC, 2023), sees substantial precipitation, resulting in approximately 345 km³ of runoff. In contrast, the dry season, spanning from December to May, witnesses a significant decrease in basin-wide precipitation, leading to a notable drop, approximately 67%, in runoff delivered to the Delta region compared to the wet season. The mainstream runoff primarily stems from recharge by upstream sub-basins, tributaries, and precipitation.

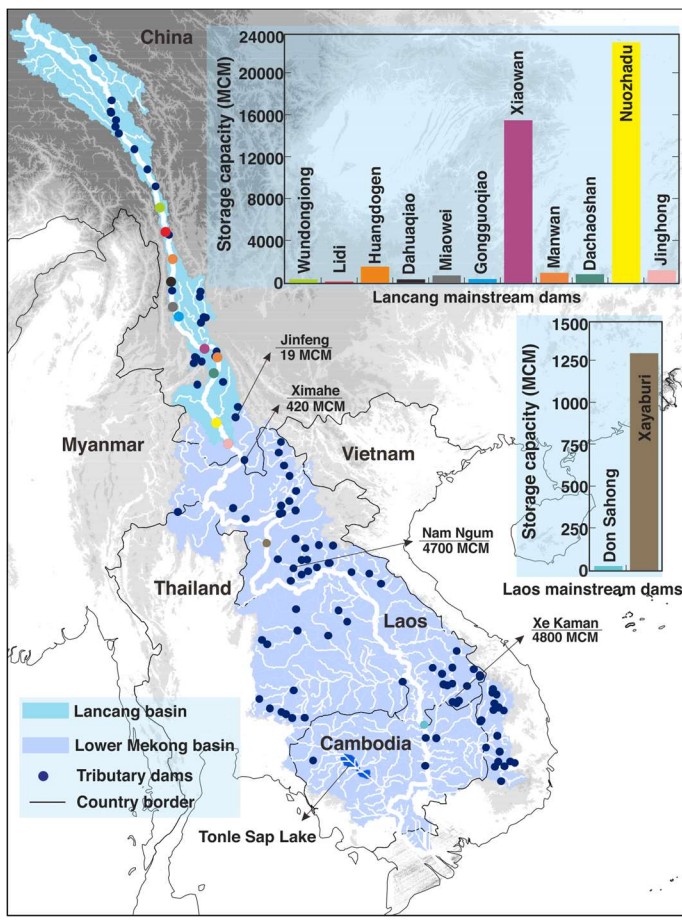


**Figure 1.** Map of the Lancang and lower Mekong basins, highlighting the extensive river network that dominates the region, along with the location of tributary and mainstream dams. The bar chart shows the total storage capacity of mainstream dams constructed within the Lancang River and lower Mekong.

## 2.2 Methodology and Data Collection

The daily fluctuations in river flow are analyzed at seven mainstream gauging stations, a process pivotal in the formulation of our sub-basin modeling framework. Each sub-basin's delineated area precisely reflects the geographical extent influencing its respective downstream station. Our developed hydrological model, as detailed in sub-sections 2.2.2 and 3.1.1, demonstrates the capability to generate time series discharge data for both mainstream stations and tributaries. These datasets serve as crucial input discharge data for defining the inlet boundary, complemented by outlet boundary specifications
derived from water level data sourced from the Mekong River Commission (MRC). This facilitates the integration of our hydrodynamic and response time models, as depicted in Figure 2.

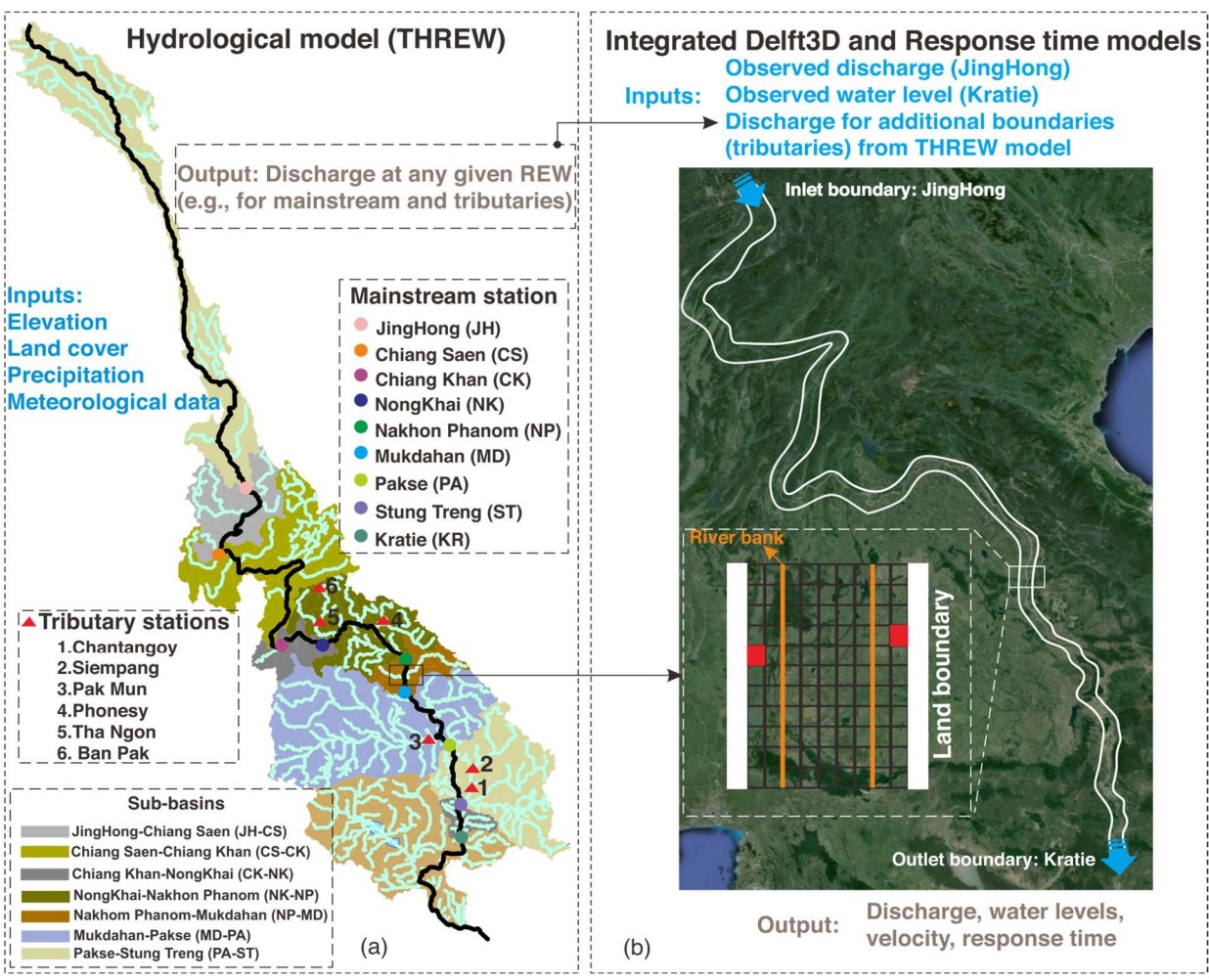

**Figure 2. Illustration of developed integrated modeling framework: (a) the THREW hydrological model applied to the LMR basin: (b) the defined computational domain (white splines, i.e., land boundary) in the developed hydrodynamic model for analyzing daily river flow fluctuations. Each tributary is represented by a single cell located between the land boundary and the riverbank (red cells in panel (b)). Note: the cells in panel (b) do not represent the actual number of cells used in the simulations (see Section 4 of the SM file for more details). The name of each defined sub-basin in this study is based on its upstream and downstream stations (panel (a)). The background map in Figure 2b is adapted from Google Earth Maps (e.g., © Google Maps).**

### 2.2.1 Data

For eight gauging stations extending from Chiang Saen (CS) to Kratie (KR) stations, continuous daily water level, and discharge data were obtained from the MRC website for the period 1980 to 2020. For tributaries, low-resolution discharge data was accessible for six stations: Chantangoy, Siempang, Pak Mun, Phonesy, Tha Ngon, and Ban Pak Kanhoung. The locations of these stations are indicated in Figure 2. Additionally, low temporal resolution velocity data were measured at the Stung Treng (ST) station; however, velocity data were not available for stations located upstream of Stung Treng.

Meteorological data sourced from the MRC and CMA, i.e., China Meteorological Administration, served as primary inputs for the THREW hydrological model. Daily precipitation records were gathered from 89 meteorological stations for the lower Mekong sub-region and 11 stations for the Lancang sub-region (see Figure S6). Additionally, other meteorological data including near-surface air pressure, air temperature, specific humidity, wind speed and direction, sunshine duration, and solar radiation were collected from 44 stations in both sub-regions. These datasets were utilized to calculate potential evapotranspiration using the Penman-Monteith Equation, which is a crucial parameter for the THREW model.

Soil data were obtained using the global soil database provided by the Food and Agriculture Organization of the United Nations (FAO), with a spatial resolution of $10 \times 10$ km. DEM data used for the THREW model were obtained from SRTM (Shuttle Radar Topography Mission), with a spatial resolution of 250 m. Data for the Normalized Difference Vegetation Index (NDVI), Leaf Area Index (LAI), and snow cover were sourced from MODIS, featuring a spatial resolution of $500 \times 500$ m and a temporal resolution of 16 days (Zhang et al., 2023).

For the developed hydrodynamic model, SRTM data with an original resolution of 90 m were utilized for areas outside the mainstream of the LMR (https://srtm.csi.cgiar.org/srtmdata/). For the mainstream of the LMR, measured cross-sectional shapes were available at various stations from the MRC website. An anisotropy approach was adopted during depth interpolation due to its superior performance in a flow-oriented coordinate system (see Merwade et al., 2006). The bathymetry data were then interpolated using the triangular technique embedded in the Delft3D model. Additionally, the internal diffusion method was applied to non-interpolated parts to assign depths to these areas (see Deltares, 2014 for detailed information). More details can be found in Section 3 of the SM file.

## 2.2.2 Hydrological Model

THREW—the Tsinghua Hydrological model based on the Representative Elemental Watershed (REW)—model serves as a physically and spatially distributed model that utilizes the REW method. This approach's spatial feature allows each REW to be divided into various hydrological zones, capturing the basin's heterogeneous nature. The model incorporates various hydrological processes, such as glacier and permafrost dynamics, snowmelt, and precipitation, making it applicable to various regions within the LMR basin. It demonstrates high performance in producing tributary flows in each REW (Cui et al., 2023). Based on the REW method, we divided the LMR basin into sub-basins based on selected hydrological stations for this study (see Figure 2).

The LMR basin is covered by 651 REWs (Figure S7). For site-based data, the Thiessen Polygon method was employed to calculate inputs for each REW. For raster data, such as LAI and NDVI, we conducted spatial intersection analysis to determine the raster cells within each REW and their respective weights. These weighted values were then averaged to obtain the inputs for the respective REW. Calibration of the model was achieved using an automatic parallel computation program to adjust hydrological parameters (Nan et al., 2021). The THREW model has been successfully applied to large river basins such as the LMR basin (Tian et al., 2020; Morovati et al., 2023), the Urumqi River basin (Mou et al., 2009), the Han River basin (Sun et al., 2014), and Yarlung Tsangpo-Brahmaputra River basin (Xu et al., 2019; Nan et al., 2021; Cui et al., 2023). Additionally, inundation is calculated using a hydrodynamic model, with the THREW model providing tributary streamflow as inputs, rather than simply spreading the runoff across the basin.

### 2.2.2.1 Reservoir Module

The THREW model schedules reservoirs according to the REW format. Due to the unavailability of detailed dam attributes, the model considers 85 dams within the basin, a number similar to that reported by Shin et al., (2020) and Dang et al., (2022). The basin contains 651 REWs (Figure S7) and each dam is assigned to its corresponding REW based on location information. For each REW, the annual cumulative reservoir storage is calculated and input as a parameter into the THREW model. The reservoir module consists of 2 parts: (1) the initial storage phase and (2) the normal operation phase. In phase 1, each REW experiences a change in cumulative storage annually, signifying the operation of new reservoirs within that REW during that year. The rules governing the initial storage phase are detailed in Equations (1) to (6). During this phase, if the inlet flow is below the minimum reservoir discharge constraint, the outlet flow equals the inlet flow. Conversely, when the incoming flow meets or exceeds the minimum reservoir discharge constraint, the outlet flow is set to this minimum value. Additionally, once the reservoir storage surpasses the minimum reservoir storage constraint, the initial storage phase concludes, transitioning the reservoir scheduling into the normal operation phase.

$$Q_{out} = \begin{cases} Q_{in}, Q_{in} < Q_{min} \\ Q_{min}, Q_{in} \geq Q_{min} \end{cases} \tag{1}$$

$$S_t = S_{t-1} + Q_{in} - Q_{out} \tag{2}$$

$$S_0 = 0 \tag{3}$$

$$if \ S_t \geq S_{min}, break \tag{4}$$

$$S_{min} = 0.2 \times S_{total} \tag{5}$$

$$Q_{min} = 0.6 \times Q_{ave} \tag{6}$$

Where $Q_{out}$ represents the outlet flow, $Q_{in}$ denote the inlet flow, $Q_{min}$ is the minimum reservoir discharge constraint, $S_t$ stands for reservoir storage at time t, $S_{min}$ is the minimum reservoir storage constraint, $S_{total}$ denotes the total reservoir storage, and $Q_{ave}$ denotes the average multi-year runoff for each REW during the calibration period (i.e., 2000-2009).

The scheduling rule for the normal operation phase of the reservoir follows the improved Standard Operation Policy hedging model (SOP rule) (Morris and Fan, 1998; Wang et al., 2017b). During this phase, the reservoir operates according to the following rules, prioritized in decreasing order from (a) to (e):

(a). Water balance: $S_t = S_{t-1} + Q_{in} - Q_{out}$

(b). Reservoir storage constraint: $S_{min} \leq S_t \leq S_{max}$

(c). Reservoir discharge constraint: $Q_{min} \leq Q_{out} \leq Q_{max}$

(d). Reservoir storage is maintained at $S_c$ in the wet season

(e). Reservoir storage is maintained at $S_n$ in the dry season

Where $S_c$ represents the reservoir storage corresponding to the flood control level and $S_n$ denotes the reservoir storage corresponding to the normal storage level.

During the normal phase, the reservoir scheduling rules account for two scenarios: the general case and the emergency case, each with distinct constraints. If, after scheduling based on the general case constraints, the outlet flow fails to meet the maximum or minimum reservoir flow constraints, the situation is deemed a contingency case. In such instances, the reservoir is re-scheduled according to the emergency case constraints, which involve appropriately relaxing the constraints on maximum reservoir storage and minimum reservoir flow. This adjustment aims to mitigate excessively high or low outlet flows, thereby reducing flow variability. While ensuring reservoir storage remains safe, the emergency case maximizes the reservoir's regulation capabilities to promote more favorable downstream ecological conditions and support downstream production and livelihoods. The reservoir scheduling rules for the emergency case are denoted by rules (f) and (g).

(f). After scheduling, verify whether the outlet flow $Q'_{out}$ is maintained between $Q_{min}$ and $Q_{max}$: $Q_{min} \leq Q'_{out} \leq Q_{max}$

(g). If this condition $f$ is false, repeat steps (a) to (e).

According to Tennant (1976), 30% of the average multi-year flow sustains good survival conditions for most aquatic life forms and basic recreation, while 10% supports the short-term survival of aquatic life forms, and 60% provides excellent habitat during their primary growth period and for recreational uses. The maximum flow released from the dam should not exceed twice the average flow (Tennant, 1976). Therefore, in the general case, $Q_{max} = 2 \times Q_{ave}$, $Q_{min} = 0.6 \times Q_{ave}$. In emergency case, $Q_{min} = 0.3 \times Q_{ave}$. $Q_{ave}$.

Referring to Yun et al., (2020) for $S_c$ and $S_n$, we set $S_c = S_{min} \times 1.2$ and $S_n = S_{max} \times 0.8$. Here, $S_{min} = 0.2 \times S_{total}$. Under the general case, $S_{max}$ varies seasonally as follows:

$$S_{max} = \begin{cases} 0.8 \times S_{total}, \text{month} = 6,7,8,9,10 \\ 1 \times S_{total}, \text{month} = 11,12,1,2,3,4,5 \end{cases}$$

Under the emergency case, $S_{max} = 0.8 \times S_{total}$.

**2.2.3 Hydrodynamic Model**

Water level/flow modeling requires a hydrodynamic model to accurately capture the daily fluctuations in river flow. Flow velocity is equally essential in this study, as any daily change in upstream flow necessitates time for its downstream impact to manifest. The Delft3D model was selected for implementing 3D simulations of the basin (Deltares, 2014). The simulation domain encompasses the river reach between JingHong and Kratie stations (∼ 2200 km) (see Figure 2).

In the study area, the horizontal scale of the river significantly exceeds its depth, validating the shallow water assumption. The Navier-Stokes equations are solved for the river's incompressible flow. Given the curved computational boundaries typical of rivers like the LMR, a spherical coordinate was utilized to prevent discretization errors resulting from undefined rectangular cells. The cyclic method was chosen for advection, and the $k - \varepsilon$ turbulence model, known for its superior performance, was implemented for simulations (Shi et al., 2022; Morovati et al., 2021b, Wu et al., 2024). Both models have been validated to produce more accurate results than their counterparts for the LMR (Morovati et al., 2023).

Land boundaries were defined wider than the river's main channel to accurately model large increases in discharge without being impacted by land boundaries. JingHong (JH) and Kratie (KR) stations were designated as the inlet and outlet boundaries, respectively, with measured daily discharge and water level defined at these points. All tributaries were also designated as additional inlet boundaries, with their daily discharge simulated by the THREW model. A vertical uniform profile was applied for the defined inlet discharge. Further details on model settings, computational meshing domain, and mesh sensitivity analysis can be found in the SM file, Sections 1 and 4.

**2.2.4 Response Time Model**

This model is developed to determine the time required for upstream daily river flow to impact the downstream section, thereby allowing us to identify which upstream daily river flow shifts correspond to specific downstream daily fluctuations. The concept of response time reflects the degree of water exchange by depicting the residence time of water bodies while taking into account spatial heterogeneity. Therefore, to determine the response time of water and temporal dynamics within river systems, a sophisticated 3D response time (age) is developed, leveraging the hydrodynamic model within the Eulerian framework (Shi et al., 2023; Wu et al., 2024). Within this model, the trajectory of water entering the river is meticulously traced through the utilization of a virtual passive substance, commonly referred to as a tracer. Note that the defined tracer does not change the water density. The response time is how long it's been since it left a specific place, like the inlet boundary. We start counting the response time from zero when the water first leaves the inlet boundary. In this context, the term "age" signifies the average time of the tracer. This means response time is calculated by considering the ages of all individual tracers and weighting them based on their mass. To simplify, the introduction of water age concentration helps

with averaging by combining the average response time of water tracers with their concentration. This creates a single variable that represents both the response time and abundance of the tracers. Both tracer concentration and age concentration follow advection-diffusion equations, which are based on the principles of mass conservation. This means that they account for the movement and spreading of tracers in the system while ensuring that mass is conserved throughout the process. The evolution of response time is governed by a set of equations (Equations 7-9), where $a$ represents the response time, $C$ is the tracer concentration, and $\alpha$ denotes the age concentration.

$$\frac{\partial C}{\partial t} + (\nabla . \vec{u})C + \frac{\partial wC}{\partial z} = \nabla . (D_h \nabla)C + \frac{\partial}{\partial z}\left(D_v \frac{\partial C}{\partial z}\right) \tag{7}$$

$$\frac{\partial \alpha}{\partial t} + (\nabla . \vec{u})\alpha + \frac{\partial w\alpha}{\partial z} = \nabla . (D_h \nabla)\alpha + \frac{\partial}{\partial z}\left(D_v \frac{\partial \alpha}{\partial z}\right) + C \tag{8}$$

$$a = \frac{\alpha}{C} \tag{9}$$

In these equations, the components of horizontal and vertical velocity (diffusivity) are represented by $\vec{u} = (u, v)$ ($D_h$) and $w$ ($D_v$), respectively. $\nabla$ is the Hamiltonian operator ($\nabla = (\partial/\partial x, \partial/\partial y)$)

At the upstream (downstream) boundary of the LMR, designated as JingHong (JH) (Kratie (KR)), the boundary conditions for the tracer and age concentrations are defined as 1 and 0 (0 and 0), respectively. These defined boundary conditions facilitate the accurate simulation of response time along the primary flow path within the LMR. Furthermore, a cold start is implemented, initializing the entire domain with zero values for both tracer and age concentrations.

## 2.3 Scenario Setting

Our regional assessment of the basin is based on observed and simulated data spanning four decades, from 1980 to 2020. We divided this period into three distinct phases. The first phase, from 1980 to 1991, is designated as the pre-dam period due to the absence of large tributary and mainstream dams, as well as limited land cover changes aimed at improving farming practices (Chua et al., 2022, Zhang et al., 2023). The second phase, spanning from 1992 to 2009, is categorized as the growth period, marked by the commencement of dam construction, including projects like the Manwan and JingHong dams, alongside observable changes in land cover (Morovati et al., 2024). The third phase, from 2010 onwards, is termed the mega-dam period, characterized by the construction of mainstream dams with a total capacity of 45 km$^3$ and many irrigation projects (Morovati et al., 2024) Additionally, a resurgence in tributary dam construction was observed in downstream sub-basins of the Lancang River, contributing to a total capacity of around 30 km$^3$.

In line with recommendations from the MRC regarding allowable hourly water level changes downstream of cascade dams (5 cm/hour or 1.2 m/day) (MRC, 2020), our study focuses on water level changes exceeding 1 meter, referred to as 'events.' The aim is to quantitatively assess the regional impacts contributing to these events.

## 3 Results

### 3.1 Model Validation

#### 3.1.1 THREW Hydrological Model

The model calibration utilized data spanning from 2000 to 2009 across all selected stations, ensuring its accuracy and reliability. The model validation was conducted for pre-dam (Figure S8) and mega-dam periods (Figure 3) to further assess its performance. The model exhibited good performance across all stations, consistently achieving an average Nash-Sutcliffe Efficiency (NSE) greater than 0.92 for pre-dam and 0.78 for mega-dam periods.

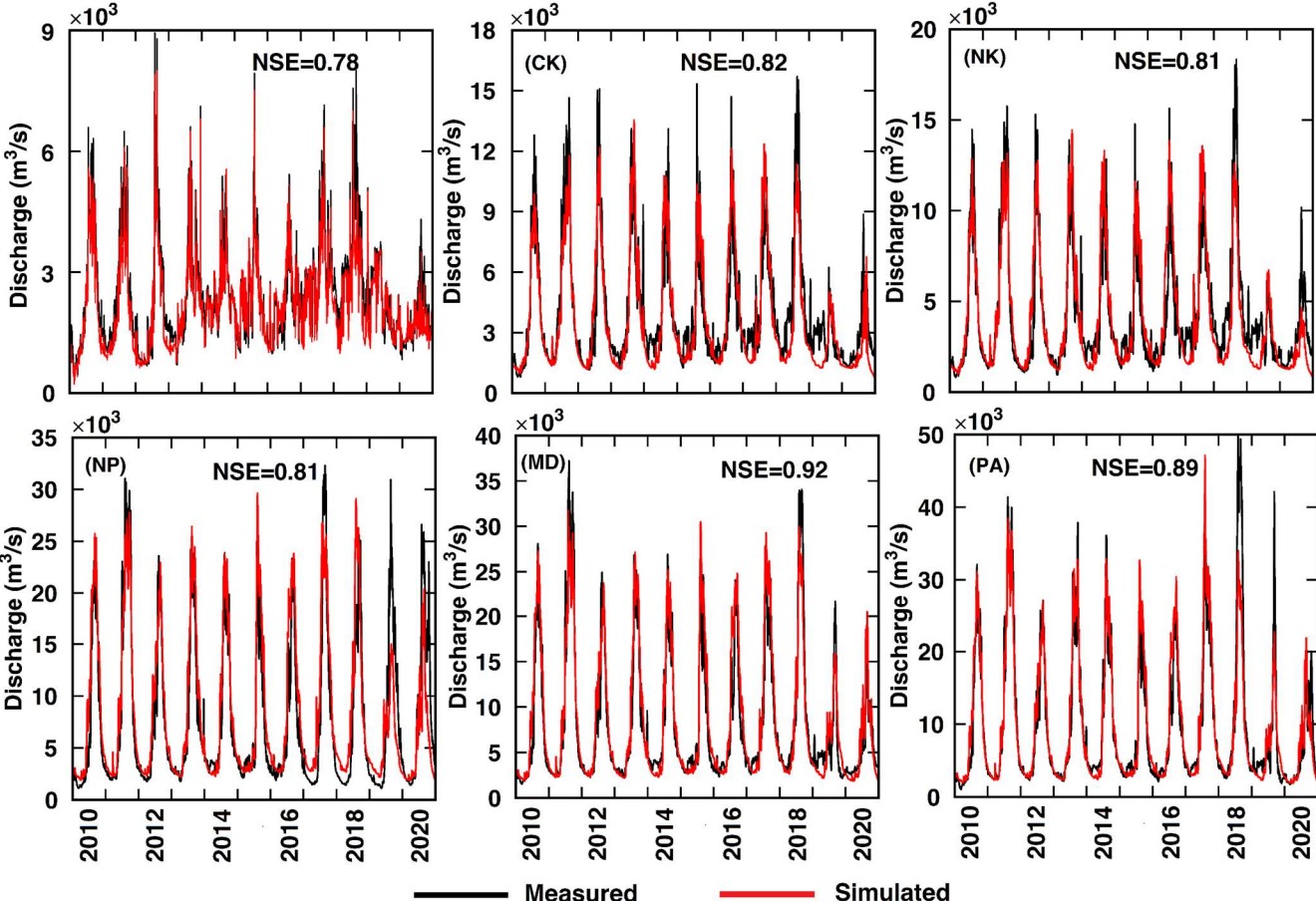

**Figure 3. Comparison of the simulated time series discharge by the THREW hydrological model with measured data over the mega-dam period (2010-2020).**

Comparable performance was attained for available tributary discharges during the pre-dam, growth, and mega-dam periods, with the NSE values exceeding 0.88, indicating high accuracy. Additional details can be found in Figure S9 of the SM file.

**3.1.2 Hydrodynamic Model**

Accurately capturing the daily large fluctuations stands as a primary objective of this study, given its significant impact on regional contribution analysis. While a comprehensive comparison of the time series discharge and water level data yielded by the hydrodynamic model is conducted for all stations throughout the study period (Figure S5), Figures 4a and b illustrate water level and discharge profiles for a single month, showcasing notable river flow shifts at Chiang Khan (CK) and Pakse (PA) stations, respectively.

In Figure 4a, observations over the span of a month reveal two substantial daily water level increases (1.5 m and 2.18 m) and a 1 m decrease in water level. Meanwhile, Figure 4b depicts Pakse station experiencing three consecutive large daily water level/flow increases. Notably, the developed model adeptly captures the river flow profiles at both stations, with a mean relative error (MRE) of less than 5%, underscoring its accuracy in modeling daily water level/flow shifts in the LMR. Regarding flow velocity, data are solely available for the Stung Treng (ST) station with low temporal resolution. Daily flow

velocity is compared with the model-derived velocity for the year 2020. A detailed point-by-point comparison indicates the model's relatively accurate simulation of flow velocity at this station, with an MRE of less than 6.2%. Comparable levels of accuracy are achieved for the years 2018 and 2019 (Figure S4).

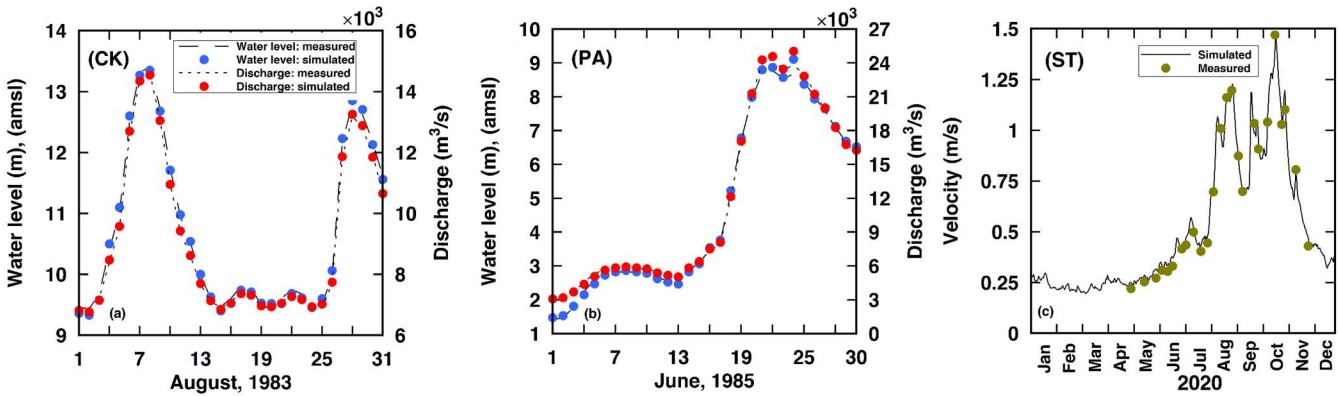

**Figure 4. Comparison of water level and discharge profiles at two stations: (a) Chiang Khan station and (b) Pakse station. Part (c)**
**depicts a point-by-point comparison of measured velocity with the hydrodynamic model for the year 2020 at the Stung Treng (ST) station.**

**3.2 Large daily water level/flow changes**

Figure 5 illustrates significant fluctuations in water levels/discharges over a 24-hour cycle (daily) across all main hydrological stations during the pre-dam period (1980-1991). For comprehensive data covering the growth and mega-dam

periods, please refer to the SM file, Figure S10. Such large river flow fluctuations occurred in the basin even before the construction of dams. During this period, a total of 143 events were recorded at these stations, with a notable concentration within the initial three months of the wet season, spanning from June to August. The number of daily fluctuations varies among the stations along the LMR. For instance, while the Pakse (PA) station encountered 28 events, its downstream station

Stung Treng (ST) experienced only 7 events. This discrepancy underscores the significant influence of regional contributions on exacerbating or mitigating downstream discharge and subsequent water level changes (see Figure 7). Furthermore, the majority of these events were characterized by substantial increases in water level/discharge, aligning with observations typically associated with the wet season, spanning from June to November.

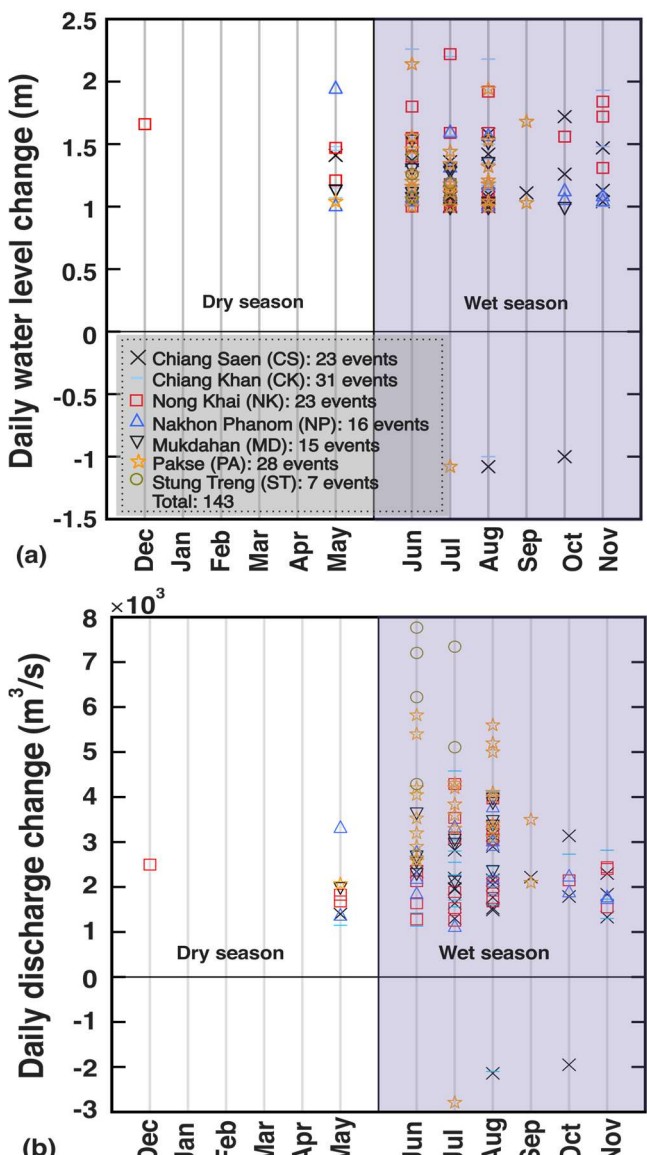

**Figure 5. The daily river flow alterations greater than 1 m for the mainstream stations (pre-dam period). Negative and positive values denote decreases and increases in water levels (a) / discharges (b), respectively. Note: in this study, the wet season starts on June 1st and ends at the end of November (LMC and MRC, 2023).**

### 3.3 Response Time

Response time denotes the duration required for a daily flow change upstream to propagate and be recorded at a downstream
gauging station. This allows us to determine the effects of upstream daily river flow changes on downstream shifts. Figure 6
illustrates the resultant graphs and their corresponding equations for each mainstream station, enabling the calculation of
response time for daily river flow changes to their respective downstream stations. Generally, a higher discharge at a given
station corresponds to a shorter response time to its downstream station. This graph facilitates the determination of the
minimum and maximum response time range for each daily river flow to reach its downstream station. For instance, at the
Pakse station (PA), situated approximately 200 km upstream from the Stung Treng station (ST), the response times range
from 1 to 10 days depending on the discharge at the Pakse station.

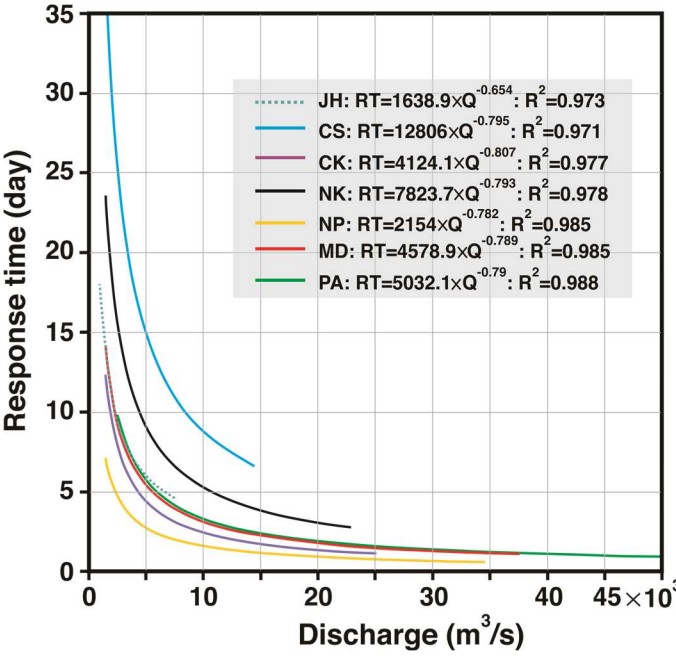

**Figure 6. Response times (RT) equations and their corresponding graphs for all mainstream stations. Please note that the results presented in this figure are derived from the developed hydrodynamic model. These equations only calculate the response times based on the upstream station and do not consider the tributaries flowing into the mainstream**

### 3.4 Contribution of sub-basin to mainstream flow

Figure 7 provides data on the cumulative average discharge at each hydrological station along the mainstream, reflecting
contributions from both the respective sub-basin and its upstream station. Upon analysis, it becomes evident that most sub-
basins in the LMR significantly contribute to the total downstream runoff, with the exception of the NP-MD and CK-NK
sub-basins. In these cases, the contribution from each respective sub-basin to its downstream station is less than 5% of the
total discharge passing through each station. On average, around 35%, 46%, and 45% of the total discharge during the wet

season passing through Chiang Saen (CS), Chiang Khan (CK), and Nakhon Phanom (NP) stations originates from respective sub-basins (Figure 7), indicating the significant role of these sub-basins in the daily river flow changes at their downstream stations.

Throughout the examined periods, there is no notable variance in the contribution of sub-basins and upstream stations to the downstream stations, except for the Nakhon Phanom (NP) station. At NP, a discernible increase of 10% in total runoff is observed in the recent decade compared to the pre-dam period (Figure S10, amounting to ~2500 m$^3$/s).

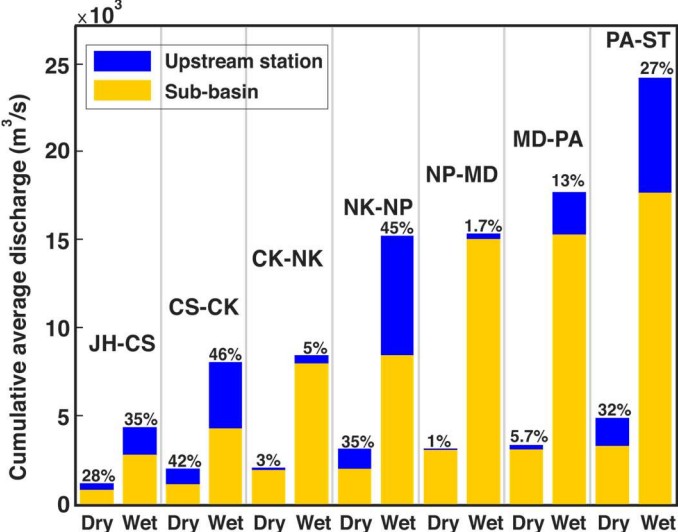

**Figure 7. The cumulative average discharge data at each hydrological station along the river's mainstream during wet and dry**
**seasons for the pre-dam period, incorporating contributions from both the corresponding sub-basin and its upstream station. Note: For example, in this figure, 'JH-CS' refers to the area/sub-basin influencing the discharge at the Chiang Saen station from the JingHong station (JH).**

### 3.5 Contribution of upstream sub-basins to large daily water level/flow increases

Figure 8 illustrates cumulative average daily discharge increases corresponding to the daily water level shifts exceeding 1m
at each station along the LMR's mainstream, considering contributions from both the respective sub-basin and its upstream station(s). Chiang Saen (CS) station, situated closest to the Lancang River course, where Chinese mega-dams were recently constructed (2010-2020). Results reveal that during the pre-dam period, 67% of the Chiang Saen's discharge that resulted in water level shifts exceeding 1m can be attributed to its respective sub-basin. However, this contribution decreased to 62% during the growth period and further to 58% during to mega-dam period, indicating an impact that surpasses human
activities in the Lancang basin.

This trend persists across other stations such as Chiang Khan (CK), Nakhon Phanom (NP), and Pakse (PA), where their respective sub-basins' contributions remained above 54% to daily discharge increases resulting in water level shifts exceeding 1m. The Mukdahan (MD) sub-basin exhibited the lowest contribution during the pre-dam period, at 9%. However,

during the growth and mega-dam periods, the average contribution to daily discharge increases surged by 28% and 55%, respectively. Conversely, at the Nong Khai (NK) station, the contribution of its sub-basin to discharge increases saw a notable decline from 33% in the pre-dam period to 17% and 4% during the growth and mega-dam periods, respectively. Notably, the Nakhon Phanom (NP) sub-basin stands out for its substantial contribution, producing 66% (pre-dam), 82% (growth period), and 86% (mega-dam) of its downstream station's large daily discharge increases.

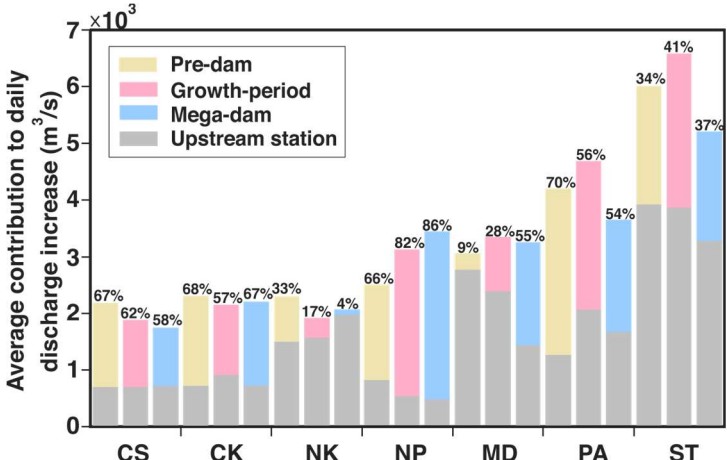

Figure 8. Average daily discharge increases corresponding to the daily water level shifts exceeding 1m at each station, considering contributions from both the respective sub-basin (colored parts) and its upstream station(s) (grey parts) for three defined periods. Percentage (%) shows the average contribution of each sub-basin in three examined periods.

## 4 Discussion

While minor alterations in the flow patterns of large rivers are expected, particularly in regions characterized by dominant tropical monsoonal climates and a high number of dam constructions, substantial shifts in river flow and water levels resulting from heavy downpours and human activities can pose significant threats to the overall integrity of river networks and subsequent aquatic productivity. This study investigated the significant changes in river flow within the recently dammed LMR basin. The analysis unveiled that, in the naturally wet conditions of the tropical lower Mekong basin, the basin experienced noteworthy large daily river flow and water level fluctuations (> 1m) even before the proliferation of anthropogenic activities such as large mainstream dams, tributary dams, and agricultural projects.

The basin's significant river flow changes primarily stemmed from increases rather than reductions in river flow, with the majority of these events occurring during the wet season over the past four decades. An approximate estimation of the precipitation received during the respective travel periods for each event reveals a consistency between the received precipitation and the contribution of the sub-basin to its downstream station during the pre-dam period, as illustrated in Figure 9. Conversely, the cumulative precipitation observed during the growth and mega-dam periods highlights a discrepancy when comparing some events: despite the larger precipitation received by the sub-basin during certain event

response times, its contribution to downstream river flow changes was less than that of events with lower precipitation (events marked by green arrows). This phenomenon could be attributed to factors such as human activities and precipitation, as indicated by the marked events in Figure 9.

Upon daily analysis of large river flow changes along the LMR, this study finds that regional assessment under a large-scale modeling framework can be observed as an effective approach rather than solely focusing on sub-basin study, as the impacts of upstream sub-basins are experienced by downstream sub-basin(s). This would also provide the possibility of exacerbating the impacts from upstream regions through coordinated management.

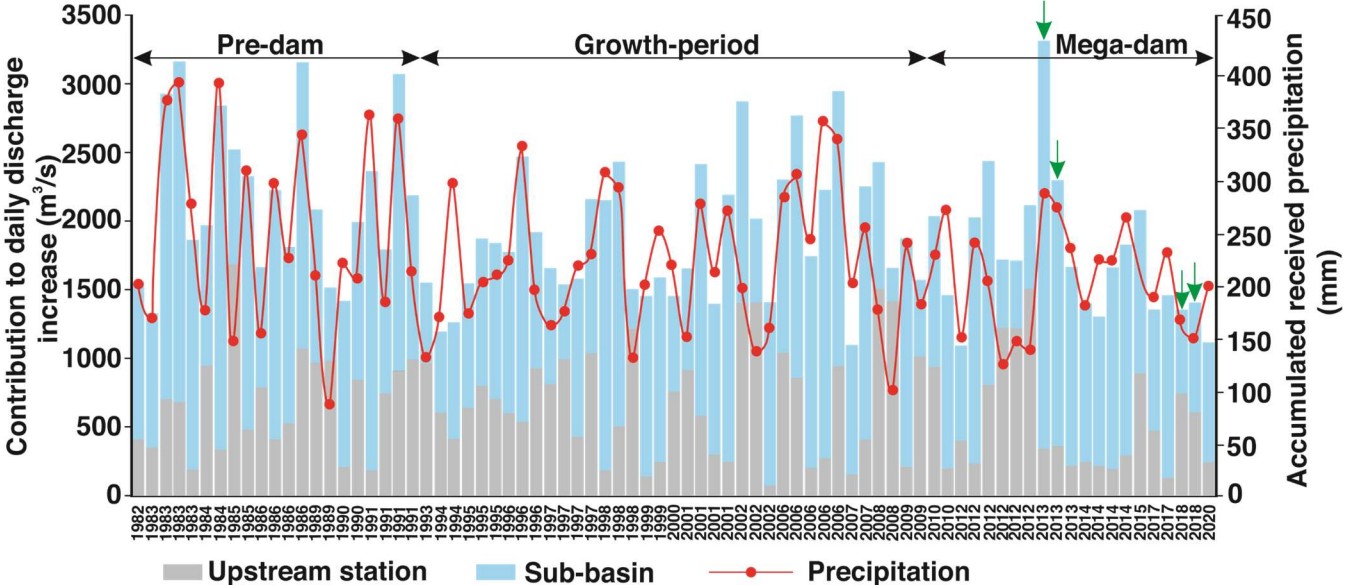

**Figure 9. Daily large river flow discharge changes in Chiang Saen (CS) station for the three defined periods. The right y-axis shows the accumulated received precipitation in the JingHong-Chiang Saen sub-basin (JH-CS). Note: The total precipitation received does not precisely correlate with the discharge corresponding to the sub-basin contribution.**

The basin experiences a notably heightened frequency of events resulting in a 1m increase in water level compared to reduction events, as illustrated by Figures 9 and 10. For example, at the CS station, there are 79 increased events while only

18 reduction events in the last four decades for all mainstream stations. This trend could possibly be due to a prevailing pattern of increased precipitation over multiple successive days, a phenomenon recurrent during the wet season (Figure S13), and previously wet conditions of the region. Particularly noteworthy is the Lancang River's significant influence on daily discharge reduction at the Chiang Saen station (CS), accounting for 8 events. This influence is especially pronounced during the growth and mega-dam periods, with more than 66% of the CS station's large river flow change attributed to the Lancang

region. This trend may be attributed to the compounded effects of heavy precipitations and human activities, such as the construction of large dams and agricultural projects after 1991.

An approximate estimation of precipitation drop – one of the indicators of climate change – before and after the response time for the JH-CS sub-basin indicates that the greater the reduction in precipitation, the higher the contribution of the sub-basin to downstream river flow decrease.

For stations like Nakhon Phanom (NP), where significant tributaries converge with the mainstream in their sub-basins, the contribution of the sub-basin to large discharge reduction outweighs that of the upstream station (59%). This can primarily be attributed to the presence of numerous tributary dams, agricultural activities (see Zhang et al., 2023), and the effects of climate change. These findings underscore the pivotal role of sub-basins in influencing downstream discharge and subsequent water level variations.

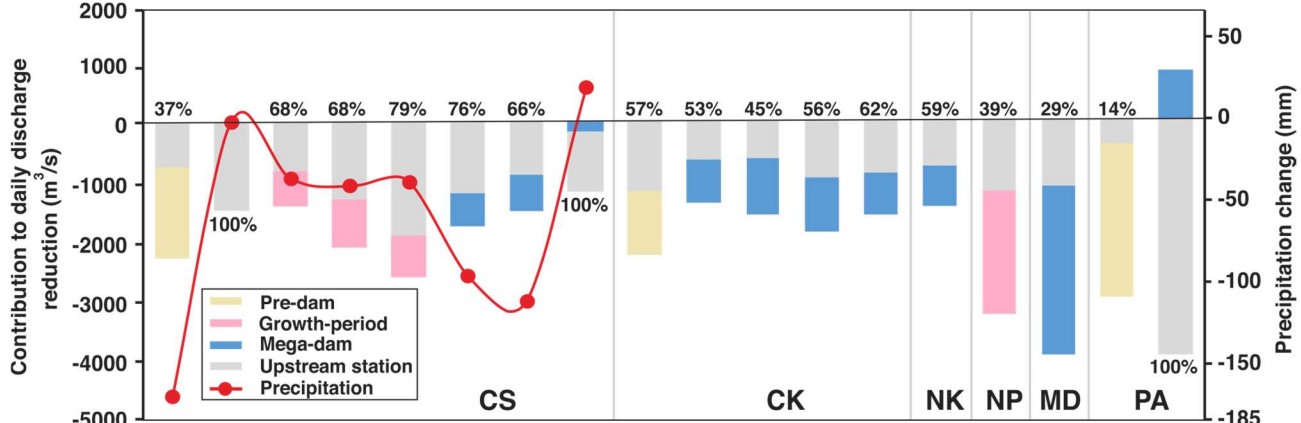


**Figure 10. The daily discharge reduction at each mainstream station, leading to water level decreases exceeding 1m, is shown based on upstream station(s) and their corresponding sub-basin. Each bar represents one event. Percentages (%) indicate the contribution of the upstream station to the downstream hydrological station's large daily discharge change. The right y-axis indicates the rough estimation of precipitation reduction before and after the response time of each event**

Any changes in the upstream regions require time to be experienced by downstream areas as river flow characteristics, including water level, discharge, and velocity are influenced. Based on the highly accurately developed hydrodynamic model, this study provided equations based on the discharge and velocity for the mainstream hydrological station (see Figure 11), this basin is lacking this data, which has great implications for future studies in terms of developing alarming systems for better management of the basin in the case of significant upstream changes in river flow as any upstream changes would

impact the flow velocity and thus response time of the impacts. These findings also bring new insights into fishery studies based on the integrated modelling frameworks, which based on our research direction, further studies will be conducted in the near future.

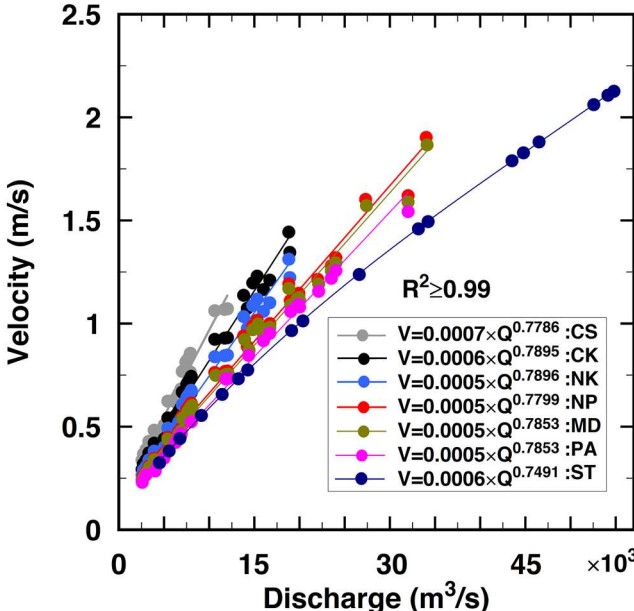

**Figure 11. Velocity equations for all mainstream stations based on the yielded data by the developed hydrodynamic model. Note:**
$R^2$ **for all equations is >0.99. Please refer to Section 10 of the SM file for more details.**

### 4.1 Limitations and Way Forward

i. This study was conducted based on the daily data as the basin lacks sub-daily data for discharge and water level. While this study provided new additional insights into the regional assessment of precipitation and human activities under a large-scale basin study, sub-daily river flow data may be central to accurately capturing the river flow regimes of the dammed LMR. For
example, at Ban Pakhoung (BP) station situated between Chiang Saen (CS) and Chiang Khan (CK) stations (see Figure 2), the reported hourly water level data reveals that fluctuations exceeding 1 m are experienced by the mainstream even within a few hours, a relatively similar pattern observes for a few days (Figure 12). This pattern can trigger fish mortality by confining fish to small water bodies, thereby reducing biomass (Li et al., 2022). This might be a result of the hourly operations of tributary dams and received precipitation by the sub-basin, as any change in its upstream station, i.e., Chiang
Saen (CS), requires more than a few days to be experienced by this station (see Figure 6).

In contrast, the water level profile does not experience fluctuations larger than 1 m when using daily data for the same period. Therefore, for the dammed basins, higher-resolution temporal data is recommended (Morovati et al., 2024), as it may help in capturing sudden water release by dams. Downscaling of the data (e.g., water level and discharge) in the LMR basin may also bring new insights into how river flow changed on an hourly scale during the pre-dam period.

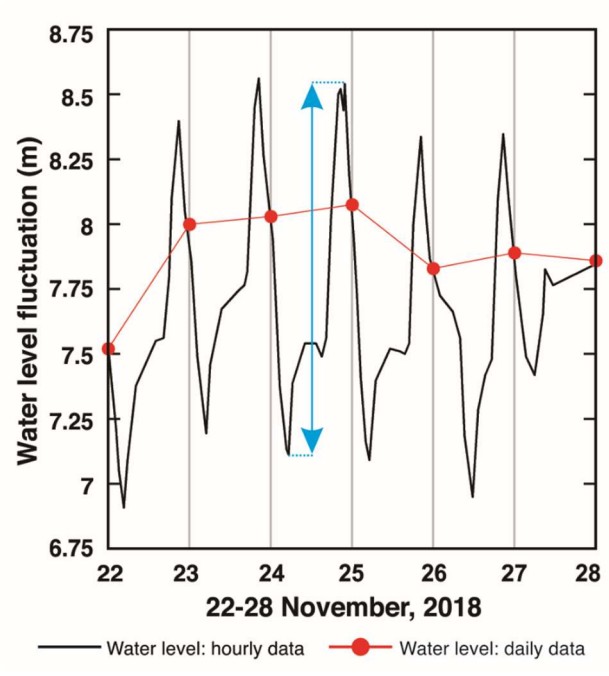


**Figure 12. Sub-daily large water level fluctuations in the Ban Pakhoung (BP) station: daily and hourly water level profiles (see Figure 2, for the location of the station).**

ii. This study was unable to explicitly address the drivers behind large flow fluctuations, a factor attributed to the lack of detailed information about the dams within the basin, including their operation rules, commissioning date, and water abstraction for agricultural purposes. Sufficient data on dam operations would enable the development of integrated hydrological, hydrodynamic, and response time models to better isolate the effects and gain a deeper understanding of the underlying mechanisms affecting water level patterns, as presented in Figure 12. The uncertainties in the data and assumptions used in the dam module (Section 2.2.2.1) may contribute to the relatively lower accuracy of the THREW hydrological model during the mega-dam period compared to the pre-dam and growth periods. Therefore, obtaining more data in these areas could potentially improve the results presented in this work.

iii. The groundwater infiltration process was not incorporated into the developed hydrodynamic model. Although its impacts seem to be insignificant compared to the large discharge passing through mainstream (Morovati et al., 2023), conducting groundwater measurements for each sub-basin, can further improve the accuracy of the model and thus reduce its uncertainty in attributing the upstream impact on downstream sub-basins.

iv. Although a large amount of sediment is transported from upstream reaches into downstream (160 Mt per year (Tian et al., 2023; Morovati et al., 2024), the basin lacks reliable sediment data to be incorporated into the model. This data is central to updating the river bed configuration while the river is simulated by the model which can negatively influence the results.

v. Low temporal resolution data for velocity was available only for the Stung Treng (ST) station within the modeled area. The developed model accurately simulates flow velocity, water level, and discharge at this station, with the same accuracy in modeling water level and discharge for the upstream mainstream station. Upon this, this study developed equations based on the river discharge and flow velocity for all mainstream stations to produce a continuous time series of velocity data. Although the accuracy in modeling flow characteristics was relatively high in mainstream stations, the accuracy of the model in reaches between two consecutive stations remains unknown, as 1) the distance between two consecutive stations is large (for example from CS to CK is around 700 km), and 2) a more turbulent flow dominates the upstream reach of the LMR compared to its downstream reach. Providing more velocity data, even with low temporal resolution, would be important in producing detailed river flow regimes for the LMR.

## 5. Conclusion

This study provided an analysis of large river flow changes across the LMR over the past four decades divided into three periods, pre-dam (1980-1991), growth (1992-2009), and mega-dam (2010-2020) periods. In doing so, a sub-basin approach was developed incorporating physically-based hydrological, reservoir module, 3D hydrodynamic, and response time models. This approach enabled us to address the contribution of sub-basins to the daily significant river flow changes. Results of response time revealed a power correlation between the upstream daily river flow changes and its required time to reach the downstream station. Daily large river flow shifts exceeding 1 m were observed in various mainstream stations even before the construction of large hydropower dams in the basin, albeit different in the number of events, emphasizing the natural variability of the river system. Approximately 92% of these significant daily water level changes occurred during the wet season, particularly in June, July, and August. These large daily river flow changes were also observed after human modifications in the basin; however, the frequency of such events did not significantly change. This study revealed that water level profiles derived from daily data (one value per day) can differ significantly from those based on hourly data (e.g., 24 values per day), potentially failing to fully capture the dynamic flow regime of the mainstream influenced by heavy downpours and the operation of numerous dams in both tributaries and the mainstream.

Moreover, we have demonstrated the substantial contribution of LMR sub-basins to mainstream discharge, with certain sub-basins contributing up to 46% to downstream mainstream stations. The JingHong-Chiang Saen sub-basin, for instance, contributed an average of 57% to significant river flow changes at the Chiang Saen station during the mega-dam period, surpassing that of the Lancang basin. This highlights the need for their consideration of basin-scale management strategies under a basin-wide approach.

## Code/Data availability

Water level, discharge, and velocity data are accessible at https://portal.mrcmekong.org/home. The model code can be obtained upon reasonable request from the primary author of the paper. Data on dams can be found in the CGIAR Research Program on Water, Land, and Ecosystems (https://wle.cgiar.org/thrive/2018/02/13/dams-data-and-decisions). SRTM data used for the hydrodynamic model can be found at https://srtm.csi.cgiar.org/srtmdata/. Due to policy restrictions, data from the JingHong (JH) station cannot be shared.

## Author contribution

KM designed the study. KM, FT, KZ, LS, and MW developed the models, with KM and KZ implementing them. KM drafted the manuscript in close collaboration with FT and YP. PS and LS contributed to data curation. Throughout the study period, all authors engaged in discussions regarding the results, provided critical feedback, and approved the final version of the paper.

## Competing interests

At least one of the co-authors is a member of the editorial board of the Hydrology and Earth System Sciences.

## Acknowledgment

This research was funded by the National Natural Science Foundation of China (NSFC) grant numbers 51825902, and 51961125204, and the Shuimu Tsinghua Scholar Program.

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
