# Peer review of "On the Cause of Large Daily River Flow Fluctuations in the Mekong River"

_Hydrology and Earth System Sciences, 2024_

## Referee Comment (RC1)

Natural fluctuations in the river are essential to the ecosystem productivity of basins. Which has less been investigated in the dammed Mekong River Basin. In view of this, this manuscript integrated a framework consisting of hydrological model, 3D hydrodynamic model, response time to address this issue. Results show that significant fluctuations in the river's daily flow were evident before the advent of the era of human activities. Further, the sub-basins were found to significantly contribute to mainstream discharge fluctuation. Overall, this manuscript is interesting, which can attract a lot of attention from readers. However, there were still some drawbacks before it is published on this journal and were listed below for references.

Major comments:

(1) The author stated that "research on the daily assessment of large river flow alterations is limited, with most researchers focusing on monthly, seasonal, and annual scale studies." (lines 59-60), which could be hard to make readers convinced. Many studies related to the discharge or floods (especially for floods) in the Mekong River Basin focused on daily scale, such as Wang et al., 2017, Wang et al., 2021 (listed by authors as references in the manuscript), Try et al. (2020), Yun et al. (2024). The word "most" could be not proper. More importantly, there should be an overview of researches on daily assessment of river flow before stating the lack of daily assessment of large river flow alterations.

*Try, S., Tanaka, S., Tanaka, K., Sayama, T., Oeurng, C., Uk, S., ... & Han, D. (2020). Comparison of gridded precipitation datasets for rainfall-runoff and inundation modeling in the Mekong River Basin. PLoS One, 15(1), e0226814.*
*Yun, X., Song, J., Wang, J., & Bao, H. (2024). Modelling to assess the suitability of hydrological-hydrodynamic model under the hydropower development impact in the Lancang-Mekong river basin. Journal of Hydrology, 131393.*

(2) The description for data was simple. Data from seven stations extending from Chiang Saen (CS) to Kratie (KR) stations were collected, however, these stations were not clearly marked in Figure 2 (only with red circles, no name was shown). This could have an impact on readers who were not familiar with this basin. In addition, the authors said that they collect many meteorological and precipitation data, but no spatial map for these sites was shown or information for these sites was revealed. It was worthy to note that the description for meteorological and precipitation data should be placed in 2.2.1, instead in 2.2.2.

(3) The methods were described relatively simple, with many confusions left, though the supplement information also contained some basic information. Firstly, readers did not know how authors calibrated the THREW model and the Delft-3D flow model, who also did not know what the parameters and inputs for these models were. Secondly, people also did not know how authors inputted the outputs of THREW model to Delft-3D flow model. I guessed that the authors used the simulated discharge near the mainstream to input to the hydrodynamic model. More importantly, how author used the meteorological data to prepare the inputs of THREW model remained uncleared (e.g., interpolating the meteorological data from in-situ scale to gridded scale).

(4) I noticed that the author used discharge to calculated the contribution to discharge, then why the hydrodynamic model was used in this manuscript. Many studies have shown that the hydrological model can well produce the discharge upstream Kratie. The author can only used hydrological model to make analyses. By the way, I am not sure why the author analyzed the velocity, which could be not important as discharge.

(5) Delft-3D flow model is a small-scale hydrodynamic model, how could author apply this model to the large basin (i.e., Mekong River Basin).

(6) The authors used "sub-basin" and "upstream station" terms many times in Section 3. For a given station, what did "sub-basin" and "upstream station" refer to. For example, in Figure 7, what did "upstream station" and "sub-basin" refer to for "PA". Could I think the "upstream station" was the nearest upstream station for a given station.

(7) The legends in Figures 9, 10 were missing. In Figure 9, what did red line, grey and blue bars represent. In Figure 10, what did the x-axis represent, For CS, why did eight bars occur. and then what did the red line represent.

Minor comments:
(1) Line 63: Usually,the trend of discharge change is similar to that of water level. Here, I am not sure why discharge increased by 98% while the water level decreased by -1.55m.

(2) Line 88: The length of the Mekong River needs further confirmation. It seems that 4500km is not a commonly used result. According to MRC (2006), the correct value is 4800km. Further, "Mekong River constitutes the third most diverse aquatic ecosystem", what were the first and second most diverse aquatic ecosystem. Mekong River should not be the second most diverse aquatic ecosystem (just followed by Amazon River Basin)?

*MRC, 2006. Annual Flood Report 2005. Mekong River Commission, Vientiane, Lao PDR,*
*p. 82.*
(3) Lines 96-97: The authors took June-December as the wet season, while took November-May as the dry season. This was not consistent with the facts. Actually, the flood season is from June to December for Mekong River Basin, while wet season is from May to October (see Räsänen and Kummu, 2013, Wang et al., 2022 for reference)

*Räsänen, T. A., & Kummu, M. (2013). Spatiotemporal influences of ENSO on precipitation and flood pulse in the Mekong River Basin. Journal of Hydrology, 476, 154-168.*
*Wang, J., Tang, Q., Yun, X., Chen, A., Sun, S., & Yamazaki, D. (2022). Flood inundation in the Lancang-Mekong River Basin: Assessing the role of summer monsoon. Journal of Hydrology, 612, 128075.*

(4) Line 226: how authors defined the daily river flow alteration, whether the authors using the water level in the next day minus that in the current day.

---

## Referee Comment (RC2)

This study investigated the cause of the large daily flow fluctuations in the Mekong River. After reading the manuscript, I have a strong feeling that the manuscript needs to be carefully revised and reviewed. Precise and clear writing is important and sufficient to report the new findings to our scientific community. Especially for the figures, some irrelevant paragraphs and unclear descriptions would confuse the readers. The authors have done a lot of work to support their findings. But the current version still needs to be improved.

**Main comments:**

1) I concur with the previous reviewer's assessment that the author's literature review of this paper requires substantial supplementation with recent content, partICularly the modelling and simulation of a series of hydrological and hydrodynamic models conducted around the Lancang-Mekong River Basin. Given that 2024 has already commenced, the modelling conducted around the LMRB has been refined to the day or even the hour. In light of the above, it is imperative that the author conducts a comprehensive synthesis and refinement of existing research, elucidating the pivotal contributions of this study. It should be noted that these works should not only be carried out in the discussion, but also require substantial supplementation and modification of the introduction.

2) The authors hope to estimate the time it takes for large daily changes in upstream rivers to affect downstream rivers, but with the large-scale construction of reservoirs and changes in river dynamics, the results of this study may not provide the expected reference value. Similarly, the authors' claim that "three aspects extend previous research" is difficult to achieve:
a) "Quantitative assessment of the regional contribution to abnormal downstream water level/flow changes". Given that there are about 500 reservoirs in the basin, I doubt the feasibility of this vision;
b) "Quantifying the propagation of upstream river flow changes to downstream sub-basins", as above, the presence of many reservoirs has significantly altered the river propagation process. Although the impact of reservoirs on mainstream flooding during the wet season is small, it should be noted that reservoir operations dominate mainstream water level changes during the dry season in the basin, and large-scale water conservation and diversion projects on tributaries have permanently altered river dynamics in these areas.
c) Due to the lack of consideration of the reservoir impact in the model, this study may only be applicable to the LMRB before 2009, and it is difficult to provide an in-depth understanding of climate impacts. Figures 3 and 4 confirm this view. The author can only show the time series verification results before 2000, and lacks the evaluation of the model effect on the tributaries and mainstream in the middle and upper reaches after the large-scale reservoir development after 2008.

3) This study may not be applicable to current LMRB. Given that this manuscript

submitted to HESS, I am a little unsure what new insights this paper can give us regarding the LMRB, especially considering that the basin has been undergoing large-scale dam construction for 20 years. Could the authors consider looking at other areas where dam construction has not yet begun, to increase the the validity of the study?

4) It should be pointed out that the author's model can obtain such a high NSE coefficient, which is mainly due to the input of the actual streamflow of the JH station. In fact, if the JH flow data is used directly to evaluate the CS flow data without considering the confluence runoff in the JH-CS sub-basin, its NSE will reach more than 0.85. However, I can't find any description of the JH station flow in the article. Considering that the streamflow data of JH station has been publicly released by the Chinese government, it is necessary for the author to make a detailed explanation.

5) As far as I know, THREW is not a gridded distributed model, but a model for lumped confluence in small catchments. How could this driven the Delft-3D model? I can't imagine flattening the confluence generated by the lumped model on an uneven DEM and expecting it to produce adequate confluence results.

6) I was unable to open the website https://portal.mrcmekong.org/home successfully, whether using the network service from German, Japan or China. Perhaps the author could consider uploading the data to such as https://zenodo.org/ for safekeeping.

**Other comments:**

7) At line88, Firstly, the official name of this basin is the Lancang-Mekong River Basin, with upstream Lancang River and downstream Mekong River. Secondly, the length of the river claimed by the author is questionable. Finally, the number of Chinese reservoirs is more than 11 and needs further verification. Considering that the collaborators include a large number of senior Chinese experts in hydraulic research, it is unacceptable to make mistakes in these details and data.

8) In Figure 1, what is "the Tonle Sap Lak"? It is recommended that the author carefully checks for the spelling and grammatical errors in the paper, as similar situations occur frequently.

9) In Figure 2, I don't think it's a good idea to use both red circles and triangles for labeling. I can't distinguish the tributary station and the mainstream station at all. Besides, I think there should be a space separating the "Delft3D".

10) Line 220, "Comparable levels of accuracy are achieved for the years 2019 and 2020, as detailed in the SM, Section 3". My understanding is that you cannot prove the overall model usability by showing only a part. ST is located downstream and has a large main stream flow, making it less affected by reservoir operation. Therefore, using flow velocity assessment at a monthly scale during the rainy season can give

better results, but this cannot prove the applicability of the model for basin-wide flow assessment after 2010.

11) The results in Figure 8 seem to be based on the comparison between the actual observed flow and the natural flow simulated by the model, or did the authors include a simulation of reservoir operation in the model? I am not sure if I missed the part about the reservoir being set up in the model. It is recommended that the authors explain how the results were obtained.

12) In Figure 9, I don't think it's a good idea to remove the year labels on the x-axis of the time series graph, as this only makes the figure harder to understand. Also, what is "recieved rainfall"? It is recommended to avoid the use of rainfall and to use precipitation uniformly. I suggest that the author consider further detailed checks on the grammar, fonts, font size, etc. of the full text and images. The current version has too many errors.

13) In Figure 10, Same as above, what is "contrinution"? I can understand that the author has a few singular and plural errors or tense problems in the manuscript. However, repeated typing errors in important figures are unacceptable.

14) In Figure 11, Please add dots of corresponding colors on the basis of the lines in the legend, which can make the image more readable.

15) In Figures 6 and 9, I am not sure how $R^2$ is calculated, what data are used? Please explain in detail.

16) Sources of meteorological soil and vegetation DEM data used in modelling must be listed in the main text in a clear and detailed manner. Layered citations are unacceptable.

---

## Author Comment (AC2)

**Reviewer 2**

**This study investigated the cause of the large daily flow fluctuations in the Mekong River. After reading the manuscript, I have a strong feeling that the manuscript needs to be carefully revised and reviewed. Precise and clear writing is important and sufficient to report the new findings to our scientific community. Especially for the figures, some irrelevant paragraphs and unclear descriptions would confuse the readers. The authors have done a lot of work to support their findings. But the current version still needs to be improved.**

**Response:** Many thanks for your feedback and valuable comments and suggestions on our manuscript. Below please find our responses to all the comments on a point-to-point basis.

**1) I concur with the previous reviewer's assessment that the author's literature review of this paper requires substantial supplementation with recent content, particularly the modelling and simulation of a series of hydrological and hydrodynamic models conducted around the Lancang-Mekong River Basin. Given that 2024 has already commenced, the modelling conducted around the LMRB has been refined to the day or even the hour. In light of the above, it is imperative that the author conducts a comprehensive synthesis and refinement of existing research, elucidating the pivotal contributions of this study. It should be noted that these works should not only be carried out in the discussion, but also require substantial supplementation and modification of the introduction.**

**Response:** Thank you for your suggestion and comment. We agree with the reviewer's assessment and will enhance the introduction by incorporating a thorough discussion on recent hydrodynamic and hydrological modeling studies, particularly those related to flow regime analysis. This will provide a deeper context and relevance to our study. Thank you again for your valuable feedback.

**2) The authors hope to estimate the time it takes for large daily changes in upstream rivers to affect downstream rivers, but with the large-scale construction of reservoirs and changes in river dynamics, the results of this study may not provide the expected reference value. Similarly, the authors' claim that "three aspects extend previous research" is difficult to achieve:**

**Response:** Thank you for your comment. We did not state "*to estimate the time it takes for large daily changes in upstream rivers to affect downstream rivers*." Rather, we highlighted the time required for upstream daily river flow to impact the downstream section (lines 241-242). Upstream daily river flow also refers to the mainstream station(s). By "downstream section," we refer to mainstream hydrological stations, as our results were reported at these stations. We did not analyze how upstream river flows influence downstream rivers because there is one river and many tributaries. If the word "rivers" refers to tributaries, we need to clarify that this study has not focused on how downstream tributaries are influenced, which is why we used terms such as "section" and "mainstream station."

We have indeed considered dams in our developed modeling framework. Therefore, all aspects highlighted in the introduction have been addressed. The first author apologizes for the oversight in not providing sufficient information on this aspect initially. Please find our detailed response below.

a) **"Quantitative assessment of the regional contribution to abnormal downstream water level/flow changes". Given that there are about 500 reservoirs in the basin, I doubt the feasibility of this vision;**

**Response:** Thank you for your comment. We believe that achieving this vision is feasible. To the best of our knowledge and based on our database, the number of constructed dams until 2020 is not 500, as indicated by the sources (https://archive.iwmi.org/wle/thrive/2018/02/13/dams-data-and-decisions/index.html, https://wle-mekong.cgiar.org/maps/). Our database shows that 99 dams were constructed before 2020 with known commercial operation dates (COD). This number increases to 284 if we include dams with unknown COD. Most of these dams are very small tributary dams, with total storage capacities such as 0.2 MCM, 4 MCM, and 0.57 MCM, among others. There are also relatively large tributary dams, as mentioned in Figure 1, with COD before 2010. Out of the 284 dams, around 228 were constructed before 2010 (including those with unknown and known COD). With known COD, this number reduces to 48 for the years before 2010.

We already provided verification results for the years before 2009, and despite the extensive number of dams, the THREW model produced accurate discharge measurements at all mainstream stations focused on in this study (NSE > 0.9). The main reason is that the vast majority of these dams have very small total storage capacities and thus have not significantly impacted the Mekong River's runoff. Many studies have confirmed that the Mekong River remained mostly unaltered by dams before this period (Pokhrel et al., 2018; Morovati et al., 2023; Grumbine and Xu, 2011; Kummu et al., 2014; MRC, 2005, etc.).

From 2010 to 2020, many large dams were built in the Lancang course and lower Mekong mainstream and tributaries. In addition to Chinese dams, two mainstream dams were constructed by Laos, both of which have been in operation since 2019. Except for two large dams constructed in the tributaries of Laos (see Figure 1), most of these are small tributary dams with limited storage capacities.

The new results for the mega-dam period (2010-2020) also confirm that the model has produced good results for the mainstream stations focused on in this study, with NSE values exceeding 0.78. Please kindly refer to Comment 3 for further details.

Grumbine, R. E., & Xu, J. (2011). Mekong hydropower development. Science, 332(6026), 178-179.

Kummu, M., Tes, S., Yin, S., Adamson, P., Józsa, J., Koponen, J., ... & Sarkkula, J. (2014). Water balance analysis for the Tonle Sap Lake–floodplain system. Hydrological Processes, 28(4), 1722-1733.

MRC. (2005). Overview of the hydrology of the Mekong Basin. Mekong River Commission, Vientiane, November 2005;82.

Morovati, K., Tian, F., Kummu, M., Shi, L., Tudaji, M., Nakhaei, P., & Olivares, M. A. (2023). Contributions from climate variation and human activities to flow regime change of Tonle Sap Lake from 2001 to 2020. Journal of Hydrology, 616, 128800.

Pokhrel, Y., Shin, S., Lin, Z., Yamazaki, D., & Qi, J. (2018). Potential disruption of flood dynamics in the Lower Mekong River Basin due to upstream flow regulation. Scientific reports, 8(1), 1-13.

b) **"Quantifying the propagation of upstream river flow changes to downstream sub-basins", as above, the presence of many reservoirs has significantly altered the river propagation process. Although the impact of reservoirs on mainstream flooding during the wet season is small, it should be noted that reservoir operations dominate mainstream water level changes during the dry season in the basin, and large-scale water conservation and diversion projects on tributaries have permanently altered river dynamics in these areas.**

**Response:** Thank you for your insightful comment. It appears to the authors that the first part of the comment, which states "Quantifying the propagation of upstream river flow changes to downstream sub-basins," stems from a misunderstanding of lines 17 and 81. The term "downstream sub-basin" suggests a relatively large area, whereas our focus is on the mainstream sections. Therefore, we propose replacing the word "sub-basins" with "sections" for clarity. One upstream river flow change may not cause a large fluctuation at a downstream mainstream station, while in other river cross-sections, it may result in significant changes due to river geometry. Therefore, our results specifically focus on the mainstream stations for which we have sufficient daily data for both discharge and water level, as shown in Figure 2. For example, the number of events (large river flow changes exceeding 1 m) shown in Figure 5 has the potential to change in upstream and downstream cross-sections of the mainstream stations because the river cross-section changes. A comparison of Figure 9 (see comment 12 for the updated version), which is based on daily analysis, with Figure 12 confirms this.

**Regarding the second part of the reviewer's comment,** our analysis based on observed data reported by the MRC shows that large changes have occurred during the wet season, not the dry season (see Figure 5). Such results can be attributed to the compounding impacts of precipitation, dam operation, and other factors. For example, heavy rainfall in an upstream sub-basin can lead to large releases by dams toward downstream areas, and if combined with downstream precipitation, this can cause significant changes in downstream river flow. In Figure 9 (see comment 12), such results are observed in the JH-CS sub-basin where only two small tributary dams were constructed before 2020 (see Figure 1). Therefore, it is difficult to assert that dams have a small impact on the mainstream during the wet season, at least for the Mekong Basin. The authors acknowledge that dams in the basin have increased river runoff during the dry season in recent decades (monthly average, etc.). However, according to our analysis based on the MRC data shown in Figure 5, these impacts have not resulted in many large river flow fluctuations exceeding 1 m, which is the focus of this study.

c) **Due to the lack of consideration of the reservoir impact in the model, this study may only be applicable to the LMRB before 2009, and it is difficult to provide an in-depth understanding of climate impacts. Figures 3 and 4 confirm this view. The author can only show the time series verification results before 2000, and lacks the evaluation of the model effect on the tributaries and mainstream in the middle and upper reaches after the large-scale reservoir development after 2008.**

**Response:** Thank you for emphasizing the need for additional details to enhance clarity for the community. We acknowledge that the current study incorporates dams in the THREW model based on available databases, as noted in Comment 3.

**Regarding the second part of the comment stating** that "we only presented verification results prior to 2000. The authors confirm that the verification results were also presented for the years after 2000. While, we acknowledge the limitations of our verification data, Figures 4c, S3, and S5 demonstrate our efforts to provide verification during both the growth and mega-dam periods. Specifically, in Figures 4c, S3, and S5. In Figures 4c and S3, we present velocity data for the years 2018, 2019, and 2020, which correspond to the mega-dam period, i.e., 2010-2020. These results were derived from our hydrodynamic model, incorporating data from tributaries within the developed THREW model. All tributaries were considered in the Delft3D model from JH to KR stations. These tributaries provide more than 80% of the total runoff of the Mekong River. We believe that highly accurate modeling of tributary discharge after 2010 by the THREW model can lead to such accuracy by the Delft3D model.

It should be noted that velocity data is only available for the ST station and for the years 2018, 2019, and 2020, and the entire dataset was utilized for model verification.

Figure S5 further compares discharge data produced by the THREW model with the full observed data available from the MRC website. For instance, in the Siempang River (see Figure S5), we achieved an NSE value of 0.89 for the years 2011 and 2012, representing the mega-dam period. Similarly, for other tributaries such as Pak Mun, Ban Pak Kanhoung, and Chantangoy, verification spans the growth period post-2000. These NSE values (>0.88) demonstrate that the developed THREW model reliably produces tributary discharge within our modeling framework. It should be noted that discharge data is only available for these tributaries, and the entire dataset was utilized for model verification.

**3) This study may not be applicable to current LMRB. Given that this manuscript submitted to HESS, I am a little unsure what new insights this paper can give us regarding the LMRB, especially considering that the basin has been undergoing large-scale dam construction for 20 years. Could the authors consider looking at other areas where dam construction has not yet begun, to increase the validity of the study?**

**Response:** Thank you for your comment. With respect to the first part of the reviewer's feedback, we believe that the three aspects highlighted in the introduction have been addressed. This study is the first to examine significant daily river flow changes in the Mekong and to provide a quantitative assessment of regional contributions. Our findings, such as those presented in Figure 6, offer valuable insights into the time required for upstream river flow changes to impact downstream stations. These insights can inform improved management strategies. Please see our description and new verifications for the mega-dam period below.

Overall, the THREW model schedules reservoirs according to the REW format. Due to the unavailability of detailed dam attributes, the model considers 85 dams within the Basin, a number similar to that reported by Dang et al., 2022. The basin contains 651 REWs and each dam is assigned to its corresponding REW based on location information. For each REW, the annual cumulative reservoir storage is calculated and input as a parameter into the THREW model.

The reservoir module of the THREW model consists of 2 parts: (1) the initial storage phase and (2) the normal operation phase.

(1) Initial storage phase:

Each REW experiences a change in cumulative storage annually, signifying the operation of new reservoirs within that REW during that year. The scheduling of these new reservoirs follows the initial storage phase rule.

The rules governing the initial storage phase are detailed in Equations (1) to (6). During this phase, if the inlet flow is below the minimum reservoir discharge constraint, the outlet flow equals the inlet flow. Conversely, when the incoming flow meets or exceeds the minimum reservoir discharge constraint, the outlet flow is set to this minimum value. Additionally, once the reservoir storage surpasses the minimum reservoir storage constraint, the initial storage phase concludes, transitioning the reservoir scheduling into the normal operation phase.

$$Q_{out} = \begin{cases} Q_{in}, Q_{in} < Q_{min} \\ Q_{min}, Q_{in} \geq Q_{min} \end{cases} \tag{1}$$

$$S_t = S_{t-1} + Q_{in} - Q_{out} \tag{2}$$

$$S_0 = 0 \tag{3}$$

$$\text{if } S_t \geq S_{min}, \text{break} \tag{4}$$

$$S_{min} = 0.2 \times S_{total} \tag{5}$$

$$Q_{min} = 0.6 \times Q_{ave} \tag{6}$$

Where $Q_{out}$ represents the outlet flow, $Q_{in}$ denote the inlet flow, $Q_{min}$ is the minimum reservoir discharge constraint, $S_t$ stands for reservoir storage at time $t$, $S_{min}$ is the minimum reservoir storage constraint, $S_{total}$ denotes the total reservoir storage, and $Q_{ave}$ denotes the average multi-year runoff for each REW during the calibration period (i.e., 2000-2009).

The scheduling rule for the normal operation phase of the reservoir follows the Improved SOP (Standard Operation Policy hedging model) rule (Wang et al., 2017; Morris & Fan, 1998). During this phase, the reservoir operates according to the following rules, prioritized in decreasing order from (a) to (e):"

a. Water balance: $S_t = S_{t-1} + Q_{in} - Q_{out}$
b. Reservoir storage constraint: $S_{min} \leq S_t \leq S_{max}$
c. Reservoir discharge constraint: $Q_{min} \leq Q_{out} \leq Q_{max}$
d. Reservoir storage is maintained at $S_c$ in the wet season
e. Reservoir storage is maintained at $S_n$ in the dry season

Where $S_c$ represents the reservoir storage corresponding to the flood control level and $S_n$ denotes the reservoir storage corresponding to the normal storage level.

In addition, the reservoir scheduling rules for the normal operation phase account for two scenarios: the general case and the emergency case, each with distinct constraints. If, after scheduling based on the general case constraints, the outlet flow fails to meet the maximum or minimum reservoir flow constraints, the situation is deemed a contingency case. In such instances, the reservoir is re-scheduled according to the emergency case constraints, which involve appropriately relaxing the constraints on maximum reservoir storage and minimum reservoir flow. This adjustment aims to mitigate excessively high or low outlet flows, thereby reducing flow variability. While ensuring

reservoir storage remains safe, the emergency case maximizes the reservoir's regulation capabilities to promote more favorable downstream ecological conditions and support downstream production and livelihoods. The reservoir scheduling rules for the emergency case are denoted by rules (f) and (g).

f.  After scheduling, verify whether the outlet flow $Q_{out}'$ is maintained between $Q_{min}$ and $Q_{max}$:
$$Q_{min} \leq Q_{out}' \leq Q_{max}$$

g.  If this condition f is false, repeat steps (a) to (e).

According to Tennant (1976), 30% of the average multi-year flow sustains good survival conditions for most aquatic life forms and basic recreation, while 10% supports the short-term survival of aquatic life forms, and 60% provides excellent habitat during their primary growth period and for recreational uses. Tennant also specified that the maximum flow released from the dam should not exceed twice the average flow. Therefore, in the general case, $Q_{max} = 2 \times Q_{ave}$, $Q_{min} = 0.6 \times Q_{ave}$. In emergency case, $Q_{min} = 0.3 \times Q_{ave}$. $Q_{ave}$.

Referring to Yun et al., (2020) for $S_c$ and $S_n$, we set $S_c = S_{min} \times 1.2$ and $S_n = S_{max} \times 0.8$. Here, $S_{min} = 0.2 \times S_{total}$. Under the general case, $S_{max}$ varies seasonally as follows:
$$S_{max} = \begin{cases} 0.8 \times S_{total}, \text{month} = 6,7,8,9,10 \\ 1 \times S_{total}, \text{month} = 11,12,1,2,3,4,5\text{'} \end{cases}$$
Under the emergency case, $S_{max} = 0.8 \times S_{total}$.

Based on this module, comparisons were conducted for the mainstream from Chiang Saen to Stung Treng, where this study has presented the results. The overall NSE of all stations is relatively high, with NSE values exceeding 0.78 (see below Figure).

[Figure]

[Figure]

**Stung Treng (g)**

**Regarding the second part of the reviewer's comment stating** "Could the authors consider looking at other areas where dam construction has not yet begun, to increase the validity of the study?

One example is the Zab sub-basin, where the Zab River is shared by Iran and Iraq. This sub-basin is one of the main contributors to the restoration of Urmia Lake, which has recently faced shrinkage due to climate change and anthropogenic stressors. Many hydraulic structures are being constructed for agricultural purposes and to divert water towards the lake.

Another example is the Mamberamo River in Indonesia, which has remained undammed.

A third example is the Salween River, shared by Myanmar, Thailand, and China.

Dang, H., Pokhrel, Y., Shin, S., Stelly, J., Ahlquist, D., & Du Bui, D. (2022). Hydrologic balance and inundation dynamics of Southeast Asia's largest inland lake altered by hydropower dams in the Mekong River basin. Science of the Total Environment, 831, 154833.

**4) It should be pointed out that the author's model can obtain such a high NSE coefficient, which is mainly due to the input of the actual streamflow of the JH station. In fact, if the JH flow data is used directly to evaluate the CS flow data without considering the confluence runoff in the JH-CS sub-basin, its NSE will reach more than 0.85. However, I can't find any description of the JH station flow in the article. Considering that the streamflow data of JH station has been publicly released by the Chinese government, it is necessary for the author to make a detailed explanation.**

**Response:** Thank you for your comment. In the modeling process, we used actual streamflow data of the JH as the inlet boundary for the hydrodynamic model and actual water level data for the outlet boundary (see Figure 2). Additionally, we defined all tributaries flowing into the mainstream within the modeling framework (Figure 2). The discharge of these tributaries was obtained using the THREW model for three defined periods.

Regarding the JH streamflow data, we regret to inform you that we are unable to make this data publicly available. We will mention this point in the revised manuscript. Thank you for your understanding.

**5) As far as I know, THREW is not a gridded distributed model, but a model for lumped confluence in small catchments. How could this driven the Delft-3D model? I can't imagine flattening the confluence generated by the lumped model on an uneven DEM and expecting it to produce adequate confluence results.**

**Response:** Thanks for your comment. The reviewer is correct that the THREW model is not a gridded distributed model. However, it is not restricted to small catchments. The THREW model has been successfully applied to large river basins such as the Urumqi River basin (Mou et al., 2009), Han River basin (Sun et al., 2014), and Yarlung Tsangpo-Brahmaputra River basin (Xu et al., 2019; Nan et al., 2021; Cui et al., 2023).

In this study, inundation is not calculated by flattening the runoff generated by the hydrological model across the basin. Instead, inundation is computed using a hydrodynamic model, with the THREW model providing streamflow of the tributaries to be used as inputs to the hydrodynamic model.

In the revised version of Figure 2 (see comment 9), the added part in the right panel shows how additional boundaries are considered in the hydrodynamic model. We mentioned in the paper that the land boundary is considered greater than the river bank. One cell was allocated to each defined tributary. This shows that the outlet discharge of the REW, which is close to the mainstream, was used to define the tributaries in the computational domain of the hydrodynamic model.

Nan, Y., Tian, L., He, Z., et al. (2021). The value of water isotope data on improving process understanding in a glacierized catchment on the Tibetan Plateau. Hydrology and Earth System Sciences. https://doi.org/10.5194/hess-2021-134

Cui, T., Li, Y., Yang, L., et al. (2023). Non-monotonic changes in Asian Water Towers' streamflow at increasing warming levels. Nature Communications, 14(1), 1176.

Sun, Y., Tian, F., Yang, L., et al. (2014). Exploring the spatial variability of contributions from climate variation and change in catchment properties to streamflow decrease in a mesoscale basin by three different methods. Journal of Hydrology, 508, 170-180.

Mou, L., Tian, F., & Hu, H. (2009). Artificial neural network model of runoff prediction in high and cold mountainous regions: A case study in the source drainage area of Urumqi River. Journal of Hydroelectric Engineering, (1), 64-69.

Xu, R., Hu, H., Tian, F., et al. (2019). Projected climate change impacts on future streamflow of the Yarlung Tsangpo-Brahmaputra River. Global and Planetary Change. https://doi.org/10.1016/j.gloplacha.2019.01.012

**6) I was unable to open the website https://portal.mrcmekong.org/home successfully, whether using the network service from German, Japan or China. Perhaps the author could consider uploading the data to such as https://zenodo.org/ for safekeeping.**

**Response:** Thank you for your comment. We have recently been informed that the MRC website was hacked, and MRC technicians are currently working to resolve the issue. This information came to our attention during recent communications with MRC and MRCS (Dr. Sarann Ly), who is a co-author of this paper.

We apologize for the oversight regarding our statement in the "Data Availability" section. While we mentioned that data are publicly available, it has come to our attention that they are accessible upon payment, after which researchers receive a license to use the data for research purposes. Due to this limitation, we are unable to upload the data:

*"In accordance with the MRC Procedures for Data and Information Exchange and Sharing of 01 November 2001, the MRC Secretariat is the Custodian of the MRC Information System. The Licensee has requested and the Licensor – MRCIS Custodian - is prepared to grant a non-exclusive, non-transferable license to the Licensee to use the Licensed Data for the purposes specified in this Noncommercial Data Use License subject to the terms and conditions contained herein".*

**7) At line 88, Firstly, the official name of this basin is the Lancang-Mekong River Basin, with upstream Lancang River and downstream Mekong River. Secondly, the length of the river claimed by the author is questionable. Finally, the number of Chinese reservoirs is more than 11 and needs further verification. Considering that the collaborators include a large number of senior Chinese experts in hydraulic research, it is unacceptable to make mistakes in these details and data.**

**Response:** Thanks for your comment. We agree with the reviewer and accordingly will use "Lancang-*Mekong River Basin*" in the manuscript.

Regarding the length of the river, we acknowledge the varying estimates in the literature. To address this, we will revise the manuscript to state that the length of the river is approximately 4800 km.

Regarding the number of large hydropower dams on the mainstream of the Lancang course until 2020, we confirm that there are 11 such dams. We understand that there may be a misunderstanding, and we apologize if the term "mainstream" was not explicitly used in line 92. We will revise the sentence to clarify that we are referring specifically to mainstream hydropower dams. However, Figure 1 and its caption indicate the presence of these 11 mainstream dams. Thank you for your feedback.

**8) In Figure 1, what is "the Tonle Sap Lak"? It is recommended that the author carefully checks for the spelling and grammatical errors in the paper, as similar situations occur frequently.**

**Response:** The first author apologizes for the oversight. It should indeed be "Tonle Sap Lake." Below is the updated version of Figure 1.

[Figure]

**9) In Figure 2, I don't think it's a good idea to use both red circles and triangles for labeling. I can't distinguish the tributary station and the mainstream station at all. Besides, I think there should be a space separating the "Delft3D".**

**Response:** Thanks for your feedback and suggestions. We use different colors for labeling as you've suggested. The updated version of Figure 2 is as follows:

Regarding the usage of "Delft3D," we followed the format used on the official website and in the user manual, where it is written as "Delft3D." We will maintain this formatting accordingly in the revised manuscript and revise line 141.

[Figure]

**Figure 2:** Illustration of developed integrated modeling framework. (a) The THREW hydrological model applied to the LMRB. (b) The defined computational domain (white splines, i.e., land boundary) in the developed hydrodynamic model for analyzing daily river flow fluctuations. Each tributary is represented by a single cell located between the land boundary and the riverbank. Note: The cells in panel (b) do not represent the actual number of cells used in the simulations. The name of each defined sub-basin in this study is based on its upstream and downstream stations (panel (a)).

**10) Line 220, "Comparable levels of accuracy are achieved for the years 2019 and 2020, as detailed in the SM, Section 3". My understanding is that you cannot prove the overall model usability by showing only a part. ST is located downstream and has a large main stream flow, making it less affected by reservoir operation. Therefore, using flow velocity assessment at a monthly scale during the rainy season can give better results, but this cannot prove the applicability of the model for basin-wide flow assessment after 2010.**

**Response:** Thank you for your comment. We stated in the manuscript that the basin suffers from insufficient data for tributaries and velocity measurements. Otherwise, we would have been able to provide more comparisons for velocity.

We believe that the ST station can be influenced by its surrounding sub-basins and upstream sub-basins. The significant river flow changes presented in Figure S8 highlight the pronounced contribution of sub-basins to these changes. The ST station is particularly influenced by the 3S basin, which is the most important tributary to the Mekong, contributing up to 20% of its flow and providing more runoff than other sub-basins. This sub-basin produces, on average, around 6600 m³/s of runoff to the ST station (see Zhang et al., 2023).

Additionally, achieving such high accuracy in velocity at this station means that the input data (e.g., from the THREW model) for an extensive number of upstream tributaries is of high accuracy; otherwise, it would be challenging to obtain such precision using low-accuracy upstream data. The developed hydrodynamic model has produced time series discharge and water levels at all mainstream stations, including the ST station, with high accuracy (NSE > 0.94).

Given these factors and the uncertainties that not only our study faces but also many other models of the Mekong basin, such as data and DEM inaccuracies, achieving MRE values of less than 7.1% based on point-by-point comparisons in different years (mega-dam period) demonstrates good accuracy and reliable modeling.

Please note that the reason we provided point-by-point comparisons for water level and discharge at other stations like Chiang Khan and Pakse (Figure 4) was to illustrate the accuracy of the developed model in simulating sharp river flow changes that occurred over consecutive days or in a short period. Our analysis is event-based rather than time series-based. Such accurate modeling of events cannot be easily observed in time series graphs like Figure 3.

Zhang, K., Morovati, K., Tian, F., Yu, L., Liu, B., & Olivares, M. A. (2023). Regional contributions of climate change and human activities to altered flow of the Lancang-mekong river. Journal of Hydrology: Regional Studies, 50, 101535.

**11) The results in Figure 8 seem to be based on the comparison between the actual observed flow and the natural flow simulated by the model, or did the authors include a simulation of reservoir operation in the model? I am not sure if I missed the part about the reservoir being set up in the model. It is recommended that the authors explain how the results were obtained.**

**Response:** Thank you for your comment. This figure does not depict comparisons between actual observed flow and natural flow simulated by the model. Instead, it presents results obtained through our developed modeling framework. We used measured data for inlet and outlet boundaries and modeled discharge using the THREW model, which accounts for approximately 130 dams in the model setup.

In the corresponding section and Figure caption, we specified that these results represent averaged outcomes of all large river flow changes. For instance, at the CS station, there were 22 events recorded from 2010 to 2020, with 58% indicating the average contribution of the sub-basin for these 22 events.

**12) In Figure 9, I don't think it's a good idea to remove the year labels on the x-axis of the time series graph, as this only makes the figure harder to understand. Also, what is "recieved rainfall"? It is recommended to avoid the use of rainfall and to use precipitation uniformly.**

**I suggest that the author consider further detailed checks on the grammar, fonts, font size, etc. of the full text and images. The current version has too many errors.**

**Response:** Thanks for your feedback and suggestions. We have added the years to the x-axis.

"Received rainfall" refers to the cumulative precipitation that the corresponding sub-basin receives during the travel time (response time). As highlighted in the paper, this value does not precisely indicate the precipitation received during the travel time but rather provides an approximate representation of the precipitation pattern during that period.

We will use "precipitation" instead of "rainfall" as suggested. Regarding the font sizes, we acknowledge that there are discrepancies among figures, and we will make specific adjustments accordingly. Figure 9 has been updated accordingly.

[Figure]

**13) In Figure 10, Same as above, what is "contrinution"? I can understand that the author has a few singular and plural errors or tense problems in the manuscript. However, repeated typing errors in important figures are unacceptable.**

**Response:** The first author apologizes for the oversight. The word "contribution" indeed should be spelled correctly. We will carefully review the manuscript to eliminate any typographical errors.

**14) In Figure 11, Please add dots of corresponding colors on the basis of the lines in the legend, which can make the image more readable.**

Response: Thanks for your suggestion. We have added the dots in the legend as recommended. Thank you.

[Figure]

**15) In Figures 6 and 9, I am not sure how $R^2$ is calculated, what data are used? Please explain in detail.**

**Response:** Thanks for your comment. We did not calculate $R^2$ for Figure 9. In Figure 6, we utilized the observed upstream discharges as inlet boundaries for the hydrodynamic model and calculated the response time for various events. Subsequently, we used the observed discharge and the derived response time to generate correlations in Excel. We determined that the "power correlation" best represents the relationship between upstream discharge and the time required for propagation to the downstream station (line 20 of the submitted manuscript). $R^2$ values were calculated for all events at each station using Excel. The same approach was followed in Figure 11.

**16) Sources of meteorological soil and vegetation DEM data used in modelling must be listed in the main text in a clear and detailed manner. Layered citations are unacceptable.**

**Response:** Thanks for the suggestion. Soil data were obtained using the global soil database provided by the Food and Agriculture Organization of the United Nations (FAO), with a spatial resolution of $10 \times 10$ km. DEM data were obtained from SRTM (Shuttle Radar Topography Mission), with a spatial resolution of 250 m.

This information will be added to the revised manuscript accordingly.

---

## Author Response (AR1)

We would like to express our thanks for the positive comments and the valuable questions/suggestions on our manuscript. We have revised the manuscript thoroughly based on all the comments. The reviewer's comments are enumerated. Our replies to each comment start with "Response."

=====================================================================

**# Editor's Comment**

**1. Thank you for the detailed replies to the very thoughtful and constructive comments of the two reviewers. I agree with the reviewers that the overall objective of your manuscript is very interesting. However, i also concur with the reviewers that, in its current form, the manuscript lacks clarity and substantial reworking is needed.**

**Response:** The authors appreciate the Editor's comment and suggestion (Professor Markus Hrachowitz) and thank the two respected reviewers for their constructive feedback. Based on the comments received, the manuscript has been significantly revised. Please find our responses to your comments and those of the reviewers below.

**2. This is in particular true for the description of the available data and, even more importantly, of the methods. At this point it remains completely unclear how the models were set up, calibrated and tested. It is also unclear how the reservoirs in the basin were accounted for, which assumptions were made and which implications these assumptions may have on the interpretation of the results.**

**Response:** Thank you for your comment and suggestion. Section 2.2.1 has been significantly revised to provide a clearer description of the data used in developing the modeling setup.

Regarding the model setup, as detailed in Section 2.2.2 and Figure 2, input data for the THREW hydrological model included DEM, land cover, precipitation, and meteorological data. Model calibration was performed using an automatic parallel computation program to adjust hydrological parameters. We have added sub-section 2.2.2.1, which describes the dam module of the THREW model developed for the basin. Please refer to sub-section 2.2.2.1 for detailed explanations.

For the Delft3D model, we used DEM, river discharge, and water level as input data. We also tested and considered various settings, including advection schemes, meshing, and turbulence models. Based on sensitivity analysis conducted for the LMR basin, the k-e turbulence model and Cyclic model were found to produce better results for flow characteristics. We have added Section 4 in the SM file to show how the hydrodynamic model has been calibrated based on the mesh sensitivity analysis, which is a key factor in obtaining reliable and accurate results. Please also refer to Section 2 of the revised manuscript for complete descriptions of models and data.

**3. Another point that is rather striking is the complete absence of any quantification of uncertainty, which is surprising and needs to be addressed.**

**Response:** Thank you for your thoughtful comment. The current work does have some uncertainties related to the data and modeling framework, which we discuss in Section 4.1. For instance, we noted that the daily data used in this study may not fully capture the dynamic flow

regime of the basin. If sub-daily data were available, the results might differ (see Figure 12). Additionally, the accuracy of the model is affected by the lack of reliable and sufficient sediment data and groundwater infiltration information (lines 453 to 459). Another significant limitation is the incomplete data on the total number of dams constructed in the basin and their attributes including the commissioned date. These factors hinder our ability to clearly isolate the impacts of precipitation, dams, or other human activities, such as agricultural practices, on the substantial daily river flow changes. Based on your feedback, we have added more details in lines 446 to 452. Thank you for your suggestion.

**Reviewer 1**

**Natural fluctuations in the river are essential to the ecosystem productivity of basins. Which has less been investigated in the dammed Mekong River Basin. In view of this, this manuscript integrated a framework consisting of hydrological model, 3D hydrodynamic model, response time to address this issue. Results show that significant fluctuations in the river's daily flow were evident before the advent of the era of human activities. Further, the sub-basins were found to significantly contribute to mainstream discharge fluctuation. Overall, this manuscript is interesting, which can attract a lot of attention from readers. However, there were still some drawbacks before it is published on this journal and were listed below for references.**

**Response:** Many thanks for your positive feedback and valuable comments and suggestions on our manuscript. Below please find our responses to all the comments on a point-to-point basis.

**(1) The author stated that "research on the daily assessment of large river flow alterations is limited, with most researchers focusing on monthly, seasonal, and annual scale studies." (lines 59 to 60), which could be hard to make readers convinced. Many studies related to the discharge or floods (especially for floods) in the Mekong River Basin focused on daily scale, such as Wang et al., 2017, Wang et al., 2021 (listed by authors as references in the manuscript), Try et al. (2020), Yun et al. (2024). The word "most" could be not proper. More importantly, there should be an overview of researches on daily assessment of river flow before stating the lack of daily assessment of large river flow alterations.**

**Response:** Thank you for your insightful comment. We agree with the reviewer's observation that the term "most" might not be appropriate in this context. Indeed, there are several studies focusing on daily scale analysis of flooding and time series discharge analysis. Our intention was to highlight that while many studies address daily river flow changes, there is a limited focus on analyzing the significant daily fluctuations in river flow and the underlying drivers of these events. In other words, this study is an event-based analysis and does not focus on daily time series analysis. As mentioned in line 192 of the first submission, this study focuses on water level changes exceeding 1m, referred to as 'event'. The number of events may vary from 0 to several events in each year. For example, the total number of such events was found to be 143 events for the years from 1980 to 1991. To make it clearer, a discussion has been added to the revised manuscript. We kindly ask the reviewer to refer to the revised introduction. Thanks again.

**(2) The description for data was simple. Data from seven stations extending from Chiang Saen (CS) to Kratie (KR) stations were collected, however, these stations were not clearly marked in Figure 2 (only with red circles, no name was shown). This could have an impact on readers who were not familiar with this basin. In addition, the authors said that they collect many meteorological and precipitation data, but no spatial map for these sites was shown or information for these sites was revealed. It was worthy to note that the description for meteorological and precipitation data should be placed in 2.2.1, instead in 2.2.2.**

**Response:** Thanks for the comment and suggestion. We agree with the reviewer's comment. In the title of Figure 2, we mentioned that "the name of each defined sub-basin is based on its upstream

and downstream stations. For example, in the JingHong-Chiang Sean (JH-CS) sub-basin, which is distinguished by a different color, JingHong is the upstream station, and Chiang Saen is the downstream station. We have updated the legend of Figure 2 by assigning different colors to each station for clarity. Please refer to Figure 2.

Regarding the meteorological and precipitation data, we concur that a spatial map showing the data collection locations would be beneficial. Consequently, we have added spatial maps to the SM file (Figure S6)

Additionally, we agree with the suggestion to move the description of the meteorological and precipitation data from sub-section 2.2.2 to sub-section 2.2.1, together with details associated with the spatial maps showing the location of stations where meteorological data was obtained. Please kindly refer to the revised manuscript lines 130 to 152.

**(3) The methods were described relatively simple, with many confusions left, though the supplement information also contained some basic information. Firstly, readers did not know how authors calibrated the THREW model and the Delft-3D flow model, who also did not know what the parameters and inputs for these models were. Secondly, people also did not know how authors inputted the outputs of THREW model to Delft-3D flow model. I guessed that the authors used the simulated discharge near the mainstream to input to the hydrodynamic model. More importantly, how author used the meteorological data to prepare the inputs of THREW model remained uncleared (e.g., interpolating the meteorological data from in-situ scale to gridded scale).**

**Response:** Thanks for the comment. Regarding the calibration of the THREW model, we calibrated the THREW hydrological model using an automatic parallel computation program to adjust hydrological parameters (see line 130 of the submitted paper, in the revised version lines 164 and 165). The input data for the THREW model, which includes elevation, land cover, precipitation, and meteorological data, is presented in Figure 2. The output of the THREW model is daily discharge, as shown in the right panel of Figure 2. The left panel of Figure 2 illustrates that the output data of the THREW model (daily discharge) is used as input data for defining additional boundaries in the Delft3D model. This information can also be found in lines 105 to 111 of the submitted paper. In the revised version, this information is found in Lines 115 to 121 and lines 234 to 235.

For site-based data, such as potential evapotranspiration and precipitation, we employed the Thiessen Polygon method to calculate inputs for each representative watershed (REW). Please note that the THREW model has been developed based on the Representative Elemental Watershed (REW) (See line 123 of the submitted manuscript and 154 in the revised manuscript). The whole basin is covered by 651 REWs. Thus, for raster data, such as the Leaf Area Index (LAI) and Normalized Difference Vegetation Index (NDVI), we conducted spatial intersection analysis to determine the raster cells within each REW and their respective weights. These weighted values were then averaged to obtain the inputs for the respective REW. We have added these explanations to the revised manuscript for clarity in lines 161 to 170.

Regarding the Delft3D model, three datasets were used as input data: discharge, water level, and DEM data (bathymetry). This information is presented in the right panel of Figure 2. However, we omitted the explanation for the DEM data used in the Delft3D model, though it is included in the SM file. We have added more explanations for the bathymetry data in section 2.2.1, lines 146 to 152 for better clarity, which reads as follows:

*"Regarding DEM data used for the developed hydrodynamic model, SRTM data with an original resolution of 90 m were utilized for the areas outside of the mainstream of the LMR. For the LMR, measured cross-sectional shapes were available at various stations from the MRC website, and an anisotropy approach was adopted during depth interpolation due to its superior performance in the flow-oriented coordinate system (see Merwade et al., 2006). The bathymetry data were then interpolated by the triangular technique embedded in the Delft3D model. The internal diffusion method was also applied to non-interpolated parts to allocate depths to these parts (see Deltares, 2014 for detailed information). More details are found in Section 3 of the SM file."*

In addition, we expanded sections 1 and 4 of the SM file to provide further description and analysis, clarifying the modeling setup in terms of settings, methods, and, more importantly, mesh sensitivity analysis, which reads as follows:

One important factor influencing the accuracy of the hydrodynamic models is mesh size, which needs to be defined for the computational domain. Various mesh sizes were analyzed and tested to produce results that closely match the measured data. A mesh convergence assessment was conducted, evaluating different sizes for the depth, length, and width of the river. The entire river stretch from JingHong to Kratie was divided into blocks, with each block containing a specific number of cells for river width and flow direction. Table S3 presents three selected mesh configurations to achieve an optimal computational domain for the model.

**Table S1.** Information on the cell sizes used for mesh convergence assessment. '*L*' represents the number of vertical layers. Note that the same number of cells was defined for each block in the flow direction.

| Mesh number | No. of cells for the width of each block | Cell size (m) | *L* |
|:---:|:---:|:---:|:---:|
| 1 | 30 | 66.6 | 8 |
| 2 | 50 | 40 | 10 |
| 3 | 70 | 28.6 | 12 |

Figure S3 presents the results of water level and velocity comparisons among the three selected mesh configurations listed in Table S1. The observations indicate that mesh refinement enhances the accuracy of water level and velocity simulations. Specifically, for water level, the Mean Relative Error (MRE) decreases from 13.5% with Mesh 1 to 6.2% with Mesh 2. Similarly, for velocity, a point-by-point comparison shows a reduction in MRE from 17.2% with Mesh 1 to 5% with Mesh 2. However, further refinement from Mesh 2 to Mesh 3 does not yield significant improvement. Consequently, Mesh 2 is chosen for all simulations in this study. To better illustrate

how cell size influences water levels, the authors have chosen to present the water level profile results for one month.

[Figure]

**Figure S3. Mesh sensitivity analysis for water level (Mukdahan (MD) station) and velocity (Stung Treng (ST) station).**

**(4) I noticed that the author used discharge to calculated the contribution to discharge, then why the hydrodynamic model was used in this manuscript. Many studies have shown that the hydrological model can well produce the discharge upstream Kratie. The author can only used hydrological model to make analyses. By the way, I am not sure why the author analyzed the velocity, which could be not important as discharge.**

**Response:** Thank you for your nice comment. We incorporated the travel time model into the hydrodynamic model to determine the time required for daily upstream changes to be experienced at the downstream station (see section 2.2.4). When a significant daily river flow occurs at the downstream station, we cannot simply attribute this change to the discharge of the previous day at the upstream station, especially considering the large distances between stations in our study area. For instance, the distance between Chiang Saen and Chiang Khan is approximately 700 km, meaning any upstream changes require several days to propagate downstream station (see Figure 6). Therefore, the integrated hydrodynamic and response time models allow us to calculate how long it takes for upstream changes to be felt at downstream stations. This capability helps us understand the causes of significant daily flow fluctuations in the Mekong River.

One important variable that can significantly influence the accuracy of the response time is velocity. High uncertainty in velocity can lead to underestimating or overestimating the response time for daily river flow changes to propagate from the upstream to the downstream station.

**(5) Delft-3D flow model is a small-scale hydrodynamic model, how could the author apply this model to the large basin (i.e., Mekong River Basin).**

**Response:** Thank you for your comment. The study area covers the river course from JingHong to Kratie station and does not encompass the entire Mekong River Basin. The updated Figure 2 now shows which area has been modeled by the Delft3D model. Further, we have successfully

applied the Delft3D model to other regions of the Mekong, such as the stretch from Kratie station to the Tonle Sap Lake floodplain, which spans approximately 500 km and covers a larger area (km²) compared to the present study (Morovati et al., 2023). Additionally, we have successfully developed a Delft3D model for the Bohai Sea, which has an area of 77,000 km², significantly larger than the area examined in this paper. For more details, please refer to our published papers.

Morovati, K., Tian, F., Kummu, M., Shi, L., Tudaji, M., Nakhaei, P., Olivares, M. A. (2023). Contributions from climate variation and human activities to flow regime change of Tonle Sap Lake from 2001 to 2020. Journal of Hydrology, 616, 128800.

Wu, M., Sun, J., Shi, L., Guo, J., Morovati, K., Lin, B., Li, Y. (2024). Vertical water renewal and dissolved oxygen depletion in a semi-enclosed Sea. Journal of Hydrology, 131369.

**(6) The authors used "sub-basin" and "upstream station" terms many times in Section 3. For a given station, what did "sub-basin" and "upstream station" refer to. For example, in Figure 7, what did "upstream station" and "sub-basin" refer to for "PA". Could I think the "upstream station" was the nearest upstream station for a given station.**

**Response**: Thank you for your comment. A sub-basin refers to the area located between two stations, as illustrated in Figure 2 with different colors. In other words, each sub-basin is defined by its upstream and downstream stations as we mentioned this point in line 115 of the submitted version and in line 127 of the revised manuscript. For example, the MD-PA sub-basin covers the area between Mukdahan (upstream station) and PA (downstream station). Regarding PA, we realized that we had omitted the name of the sub-basin, which should be PA-ST, with PA as the upstream station and ST (Stung Treng) as the downstream station. We apologize for this oversight and have re-drawn the figure accordingly. Please refer to Figures 2 and 7. Thank you.

**(7) The legends in Figures 9, 10 were missing. In Figure 9, what did red line, grey and blue bars represent. In Figure 10, what did the x-axis represent, For CS, why did eight bars occur. and then what did the red line represent.**

**Response:** Thank you for your insightful comments, and we apologize for the oversight.

Regarding Figure 9, the grey and blue bars represent the contributions of the upstream region and sub-basin, respectively. Specifically, for this figure, which addresses significant daily river flow changes at the Chiang Saen (CS) station, the grey bars show the contribution from the area upstream of the JingHong (JH) station, while the blue bars indicate the contribution from the JH-CS sub-basin to the large daily river flows at CS. Additionally, the red line represents the precipitation received in the corresponding sub-basin. We have updated the figure caption and added a legend to enhance clarity as follows:

[Figure]

Regarding Figure 10, the x-axis represents the large daily river flow reductions at mainstream stations, with the number of such reductions varying by station. As observed, these events occur more frequently at the CS station compared to downstream stations. Each bar corresponds to a single event, i.e., one large daily river flow reduction exceeding 1 meter. To make it clear, we added this description in line 417 of the revised manuscript. The colored bars denote the contributions of defined sub-basins during each study period, while the grey bars represent the upstream station's contribution. The red line indicates the precipitation received by the JH-CS sub-basin. Based on these explanations, we have updated the legend of Figure 10 for greater clarity as follows:

[Figure]

Figure 10. The daily discharge reduction at each mainstream station, leading to water level decreases exceeding 1m, is shown based on upstream station(s) and their corresponding sub-basin. Each bar represents one event. Percentages (%) indicate the contribution of the upstream station to the downstream hydrological station's large daily discharge change. The right y-axis indicates the rough estimation of precipitation reduction before and after the response time of each event

**Minor comments:**

**(1) Line 63: Usually the trend of discharge change is similar to that of water level. Here, I am not sure why discharge increased by 98% while the water level decreased by -1.55m.**

**Response:** Thank you for your insightful comment. We agree with the reviewer's observation. The reduction in water level mentioned in line 64 pertains to the wet season, which was not specified. We have revised the sentence as follows to address this oversight (lines 40 to 42 of the revised manuscript):

*"For instance, from 2010 onward, the Chiang Saen station in Thailand, near the China border, experienced a 98% increase in monthly discharge during the dry season, while the wet season water level dropped by 1.55 meters (Lu and Chua, 2021)."*

**(2) Line 88: The length of the Mekong River needs further confirmation. It seems that 4500km is not a commonly used result. According to MRC (2006), the correct value is 4800km. Further, "Mekong River constitutes the third most diverse aquatic ecosystem", what were the first and second most diverse aquatic ecosystem. Mekong River should not be the second most diverse aquatic ecosystem (just followed by Amazon River Basin)?**

**Response:** Thanks for your comment. Regarding the river length of the Mekong River, we agree with the reviewer's comment. The correct length should be approximately 4800 km, which has been updated in the revised manuscript (line 90).

Concerning the Mekong River being the second (Campbel and Barlow, 2020) or third most diverse aquatic ecosystem, there are conflicting references. Some sources state it is the second most diverse, while others indicate it is the third (e.g., Intralawan et al., 2018). We have revised the sentence to align with the reviewer's suggestion and to be consistent with our previous publications, which reads as follows (lines 90-91 of the revised manuscript): Thank you.

With ~4800 km in length, the pan-shaped LMR constitutes the second most diverse aquatic ecosystem globally (MRC, 2011; Intralawan et al., 2019) and ranks as the eighth largest in terms of annual runoff (Sabo et al., 2017).

Campbell, I., & Barlow, C. (2020). Hydropower development and the loss of fisheries in the Mekong River Basin. Frontiers in Environmental Science, 8, 566509.

Intralawan, A., Wood, D., Frankel, R., Costanza, R., Kubiszewski, I. (2018). Tradeoff analysis between electricity generation and ecosystem services in the Lower Mekong Basin. Ecosystem Services, 30, 27-35.

**(3) Lines 96-97: The authors took June-December as the wet season, while took November-May as the dry season. This was not consistent with the facts. Actually, the flood season is from June to December for Mekong River Basin, while wet season is from May to October (see Räsänen and Kummu, 2013, Wang et al., 2022 for reference)**

**Response:** Thanks for your comment. We acknowledge that several papers define the wet season from May to October. However, in our study, we defined the wet season from June 1 to November

end and the dry season from December 1 to May end (see Figure 5 and its caption, lines 320-321). This choice was informed by current projects conducted by the Mekong River Commission (MRC) and experts from the six riparian countries, who have concluded that June to November can be considered the wet season (reference provided below).

While we understand there are different definitions of the wet season, it's important to note that our research focus does not center on monthly, yearly, or seasonal flows, and thus, our findings are not affected by the specific definition of the wet season.

Thank you for bringing this clarification to our attention.

Ngor, P. B., Oberdorff, T., Phen, C., Baehr, C., Grenouillet, G., & Lek, S. (2018). Fish assemblage responses to flow seasonality and predictability in a tropical flood pulse system. Ecosphere, 9(11), e02366.

LMC (Lancang-Mekong Water Center) and MRC. (2023). Technical Report – Phase 1 of the Joint Study on the Changing Patterns of Hydrological Conditions of the Lancang-Mekong River Basin and Adaptation Strategies. Beijing: LMC Water Center or Vientiane: MRC Secretariat. http://www.lmcwater.org.cn/cooperative_achievements/collaborative_projects/ http://www.mrcmekong.org/ publications/

**(4) Line 226: how authors defined the daily river flow alteration, whether the authors using the water level in the next day minus that in the current day.**

**Response:** Thanks for your comment. In section 2.2.1 of the manuscript, we emphasized that the water level and discharge data are recorded daily and simultaneously. Therefore, when we refer to daily river flow alterations, we mean changes that occur within a 24-hour cycle, either from one day to the next or from one day to the previous day.

**Reviewer 2**

**This study investigated the cause of the large daily flow fluctuations in the Mekong River. After reading the manuscript, I have a strong feeling that the manuscript needs to be carefully revised and reviewed. Precise and clear writing is important and sufficient to report the new findings to our scientific community. Especially for the figures, some irrelevant paragraphs and unclear descriptions would confuse the readers. The authors have done a lot of work to support their findings. But the current version still needs to be improved.**

**Response:** Many thanks for your feedback and valuable comments and suggestions on our manuscript. Below please find our responses to all the comments on a point-to-point basis.

**1) I concur with the previous reviewer's assessment that the author's literature review of this paper requires substantial supplementation with recent content, particularly the modelling and simulation of a series of hydrological and hydrodynamic models conducted around the Lancang-Mekong River Basin. Given that 2024 has already commenced, the modelling conducted around the LMRB has been refined to the day or even the hour. In light of the above, it is imperative that the author conducts a comprehensive synthesis and refinement of existing research, elucidating the pivotal contributions of this study. It should be noted that these works should not only be carried out in the discussion, but also require substantial supplementation and modification of the introduction.**

**Response:** Thank you for your suggestion and comment. We agree with the reviewer's assessment and have enhanced the introduction by incorporating a thorough discussion of recent hydrodynamic and hydrological modeling studies, particularly those related to flow regime analysis. Please refer to the revised introduction for more details.

**2) The authors hope to estimate the time it takes for large daily changes in upstream rivers to affect downstream rivers, but with the large-scale construction of reservoirs and changes in river dynamics, the results of this study may not provide the expected reference value. Similarly, the authors' claim that "three aspects extend previous research" is difficult to achieve:**

**Response:** Thank you for your comment. We did not state "*to estimate the time it takes for large daily changes in upstream rivers to affect downstream rivers*." Rather, we highlighted the time required for upstream daily river flow to impact the downstream section (lines 241-242 of the submitted manuscript and lines 239 and 240 of the revised manuscript). Upstream daily river flow also refers to the mainstream station(s). By "downstream section," we refer to mainstream hydrological stations, as our results were reported at these stations. We did not analyze how upstream river flows influence downstream rivers because there is one river and many tributaries. If the word "rivers" refers to tributaries, we need to clarify that this study has not focused on how downstream tributaries are influenced, which is why we used terms such as "section" and "mainstream station."

We have indeed considered dams in our developed modeling framework. For example, Figure 4c, Figure S3, and Figure S5 of the first submitted manuscript confirm that the modeling framework has considered dams because these figures are related to the periods of 2000 to 2009 (the growth period) and after 2010 (the mega dam period). Therefore, we believe that all aspects highlighted

in the introduction have been addressed. The first author apologizes for the oversight in not providing sufficient information on this aspect initially. Please find our detailed response below.

a) **"Quantitative assessment of the regional contribution to abnormal downstream water level/flow changes". Given that there are about 500 reservoirs in the basin, I doubt the feasibility of this vision;**

**Response:** Thank you for your comment. We believe that achieving this vision is feasible. To the best of our knowledge and based on our database, the number of constructed dams until 2020 is not 500, as indicated by the sources (https://archive.iwmi.org/wle/thrive/2018/02/13/dams-data-and-decisions/index.html, https://wle-mekong.cgiar.org/maps/). Our database shows that 99 dams were constructed before 2020 with known commercial operation dates (COD). This number increases to 284 if we include dams with unknown COD. Most of these dams are very small tributary dams, with total storage capacities such as 0.2 MCM, 4 MCM, and 0.57 MCM, among others. There are also relatively large tributary dams, as mentioned in Figure 1, with COD before 2010. Out of the 284 dams, around 228 were constructed before 2010 (including those with unknown and known COD). With known COD, this number reduces to 48 for the years before 2010.

We already provided verification results for the years before 2009, and despite the extensive number of dams, the THREW model produced accurate discharge measurements at all mainstream stations focused on in this study (NSE > 0.9) (For example figure S8 for the growth period). The main reason is that the vast majority of these dams have very small total storage capacities and thus have not significantly impacted the Mekong River's runoff. Many studies have confirmed that the Mekong River remained mostly unaltered by dams before this period (Pokhrel et al., 2018; Morovati et al., 2023; Grumbine and Xu, 2011; Kummu et al., 2014; MRC, 2005, etc.).

From 2010 to 2020, many large dams were built in the Lancang course and lower Mekong mainstream and tributaries. In addition to Chinese dams, two mainstream dams were constructed by Laos, both of which have been in operation since 2019. Except for two large dams constructed in the tributaries of Laos (see Figure 1), most of these are small tributary dams with limited storage capacities.

In the first submission, Figure 4c and Figures S3 and S5 confirm that the model produced reliable results for the mega-dam period. In Figure 4c, the velocity data were simulated using the Delft3D model, while the discharges of tributaries flowing into the mainstream were obtained from the THREW model for the years 2018, 2019, and 2020. Please note that all tributaries have been considered in the Delft3D model as additional boundaries as highlighted in Figure 2 and lines 77 and 234-235 of the revised manuscript. This indicates that the dams were accurately represented in the THREW model; otherwise, achieving such accurate results would have been very challenging.

Based on this comment we have added more details in the revised manuscript including sections 2.2.2 and 2.2.2.1.

The newly added results for the mega-dam period (2010-2020) also confirm that the model has produced good results for the mainstream stations focused on in this study, with NSE

values exceeding 0.78 (see Figure 3 of the revised manuscript). Please also kindly refer to Comment 3 for further details.

Grumbine, R. E., & Xu, J. (2011). Mekong hydropower development. Science, 332(6026), 178-179.

Kummu, M., Tes, S., Yin, S., Adamson, P., Józsa, J., Koponen, J., ... & Sarkkula, J. (2014). Water balance analysis for the Tonle Sap Lake–floodplain system. Hydrological Processes, 28(4), 1722-1733.

MRC. (2005). Overview of the hydrology of the Mekong Basin. Mekong River Commission, Vientiane, November 2005;82.

Morovati, K., Tian, F., Kummu, M., Shi, L., Tudaji, M., Nakhaei, P., & Olivares, M. A. (2023). Contributions from climate variation and human activities to flow regime change of Tonle Sap Lake from 2001 to 2020. Journal of Hydrology, 616, 128800.

Pokhrel, Y., Shin, S., Lin, Z., Yamazaki, D., & Qi, J. (2018). Potential disruption of flood dynamics in the Lower Mekong River Basin due to upstream flow regulation. Scientific reports, 8(1), 1-13.

b) **"Quantifying the propagation of upstream river flow changes to downstream sub-basins", as above, the presence of many reservoirs has significantly altered the river propagation process. Although the impact of reservoirs on mainstream flooding during the wet season is small, it should be noted that reservoir operations dominate mainstream water level changes during the dry season in the basin, and large-scale water conservation and diversion projects on tributaries have permanently altered river dynamics in these areas.**

**Response:** Thank you for your insightful comment. It appears to the authors that the first part of the comment, which states "Quantifying the propagation of upstream river flow changes to downstream sub-basins," stems from a misunderstanding of lines 17 and 81 of the submitted paper. The term "downstream sub-basin" suggests a relatively large area, whereas our focus is on the mainstream sections. Therefore, we propose replacing the word "sub-basins" with "sections" for clarity. One upstream river flow change may not cause a large fluctuation at a downstream mainstream station, while in other river cross-sections, it may result in significant changes due to river geometry. Therefore, our results specifically focus on the mainstream stations for which we have sufficient daily data for both discharge and water level, as shown in Figure 2. For example, the number of events (large river flow changes exceeding 1 m) shown in Figure 5 has the potential to change in upstream and downstream cross-sections of the mainstream stations because the river cross-section changes. A comparison of Figure 9 (see comment 12 for the updated version), which is based on daily analysis, with Figure 12 confirms this.

**Regarding the second part of the reviewer's comment,** our analysis based on observed data reported by the MRC shows that large changes have occurred during the wet season, not the dry season (see Figure 5). Such results can be attributed to the compounding impacts of precipitation, dam operation, and other factors. For example, heavy rainfall in an upstream sub-basin can lead to large releases by dams toward downstream areas, and if combined with downstream precipitation, this can cause significant changes in downstream river flow. In Figure 9 (see comment 12), such results are observed in the JH-CS sub-basin where only two small tributary dams were constructed before 2020 (see Figure 1).

Therefore, it is difficult to assert that dams have a small impact on the mainstream during the wet season, at least for the Mekong Basin. The authors acknowledge that dams in the basin have increased river runoff during the dry season in recent decades (monthly average, etc.). However, according to our analysis based on the MRC data shown in Figure 5, these impacts have not resulted in many large river flow fluctuations exceeding 1 m, which is the focus of this study.

c) **Due to the lack of consideration of the reservoir impact in the model, this study may only be applicable to the LMRB before 2009, and it is difficult to provide an in-depth understanding of climate impacts. Figures 3 and 4 confirm this view. The author can only show the time series verification results before 2000, and lacks the evaluation of the model effect on the tributaries and mainstream in the middle and upper reaches after the large-scale reservoir development after 2008.**

**Response:** Thank you for emphasizing the need for additional details to enhance clarity for the community. We acknowledge that the current study incorporates dams in the THREW model based on available databases, as noted in Comment 3.
**Regarding the second part of the comment stating** that "we only presented verification results prior to 2000. The authors confirm that the verification results were also presented for the years after 2000. While, we acknowledge the limitations of our verification data, Figures 4c, S3, and S5 demonstrate our efforts to provide verification during both the growth and mega-dam periods. Specifically, in Figures 4c, S3, and S5. In Figures 4c and S3, we present velocity data for the years 2018, 2019, and 2020, which correspond to the mega-dam period, i.e., 2010-2020. These results were derived from our hydrodynamic model, incorporating data from tributaries within the developed THREW model. All tributaries were considered in the Delft3D model from JH to KR stations. These tributaries provide more than 80% of the total runoff of the Mekong River. We believe that highly accurate modeling of tributary discharge after 2010 by the THREW model can lead to such accuracy by the Delft3D model.
It should be noted that velocity data is only available for the ST station and for the years 2018, 2019, and 2020, and the entire dataset was utilized for model verification (see lines 120-121 of the first submission).
Figure S5 further compares discharge data produced by the THREW model with the full observed data available from the MRC website. For instance, in the Siempang River (see Figure S9 of the revised SM file, in the first submission it was S5), we achieved an NSE value of 0.89 for the years 2011 and 2012, representing the mega-dam period. Similarly, for other tributaries such as Pak Mun, Ban Pak Kanhoung, and Chantangoy, verification spans the growth period post-2000. These NSE values (>0.88) demonstrate that the developed THREW model reliably produces tributary discharge within our modeling framework. It should be noted that discharge data is only available for these tributaries, and the entire dataset was utilized for model verification.

**3) This study may not be applicable to current LMRB. Given that this manuscript submitted to HESS, I am a little unsure what new insights this paper can give us regarding the LMRB, especially considering that the basin has been undergoing large-scale dam construction for**

**20 years. Could the authors consider looking at other areas where dam construction has not yet begun, to increase the validity of the study?**

**Response:** Thank you for your comment. With respect to the first part of the reviewer's feedback, we believe that the three aspects highlighted in the introduction have been addressed. This study is the first to examine significant daily river flow changes in the Mekong and to provide a quantitative assessment of regional contributions. Our findings, such as those presented in Figure 6, offer valuable insights into the time required for upstream river flow changes to impact downstream stations. These insights can inform improved management strategies. Please see our description and new verifications for the mega-dam period below. These details have been added to the revised manuscript sections 2.2.2, 2.2.2.1, and 3.1.1.

[revised manuscript text omitted]

Referring to Yun et al., (2020) for $S_c$ and $S_n$, we set $S_c = S_{min} \times 1.2$ and $S_n = S_{max} \times 0.8$. Here, $S_{min} = 0.2 \times S_{total}$. Under the general case, $S_{max}$ varies seasonally as follows:

$$S_{max} = \begin{cases} 0.8 \times S_{total}, \text{month} = 6,7,8,9,10 \\ 1 \times S_{total}, \text{month} = 11,12,1,2,3,4,5' \end{cases}$$

Under the emergency case, $S_{max} = 0.8 \times S_{total}$.

Based on this module, comparisons were conducted for the mainstream from Chiang Saen to Stung Treng, where this study has presented the results. The overall NSE of all stations is relatively high, with NSE values exceeding 0.78 (see below Figure).

[Figure]

**Regarding the second part of the reviewer's comment stating** "Could the authors consider looking at other areas where dam construction has not yet begun, to increase the validity of the study?

One example is the Zab sub-basin, where the Zab River is shared by Iran and Iraq. This sub-basin is one of the main contributors to the restoration of Urmia Lake, which has recently faced shrinkage due to climate change and anthropogenic stressors. Many hydraulic structures are being constructed for agricultural purposes and to divert water towards the lake.

Another example is the Mamberamo River in Indonesia, which has remained undammed.

A third example is the Salween River, shared by Myanmar, Thailand, and China.

Dang, H., Pokhrel, Y., Shin, S., Stelly, J., Ahlquist, D., & Du Bui, D. (2022). Hydrologic balance and inundation dynamics of Southeast Asia's largest inland lake altered by hydropower dams in the Mekong River basin. Science of the Total Environment, 831, 154833.

Shin, S., Pokhrel, Y., Yamazaki, D., Huang, X., Torbick, N., Qi, J., Nguyen, T. D.: High-resolution modeling of river-floodplain-reservoir inundation dynamics in the Mekong River Basin. Water Resources Research, 56(5), e2019WR026449, 2020.

**4) It should be pointed out that the author's model can obtain such a high NSE coefficient, which is mainly due to the input of the actual streamflow of the JH station. In fact, if the JH flow data is used directly to evaluate the CS flow data without considering the confluence runoff in the JH-CS sub-basin, its NSE will reach more than 0.85. However, I can't find any description of the JH station flow in the article. Considering that the streamflow data of JH station has been publicly released by the Chinese government, it is necessary for the author to make a detailed explanation.**

**Response:** Thank you for your comment. In the modeling process, we used actual streamflow data of the JH as the inlet boundary for the hydrodynamic model and actual water level data for the outlet boundary (see Figure 2). Additionally, we defined all tributaries flowing into the mainstream within the modeling framework (Figure 2). The discharge of these tributaries was obtained using the THREW model for three defined periods.

Regarding the JH streamflow data, we regret to inform you that we are unable to make this data publicly available. We have mentioned this point in the revised manuscript, Data Availability section. Thank you for your understanding.

**5) As far as I know, THREW is not a gridded distributed model, but a model for lumped confluence in small catchments. How could this driven the Delft-3D model? I can't imagine flattening the confluence generated by the lumped model on an uneven DEM and expecting it to produce adequate confluence results.**

**Response:** Thanks for your comment. The reviewer is correct that the THREW model is not a gridded distributed model. However, it is not restricted to small catchments. The THREW model has been successfully applied to large river basins such as the Urumqi River basin (Mou et al., 2009), Han River basin (Sun et al., 2014), and Yarlung Tsangpo-Brahmaputra River basin (Xu et al., 2019; Nan et al., 2021; Cui et al., 2023). This information was added to the revised manuscript lines 168 to 170.

In this study, inundation is not calculated by flattening the runoff generated by the hydrological model across the basin. Instead, inundation is computed using a hydrodynamic model, with the THREW model providing streamflow of the tributaries to be used as inputs to the hydrodynamic model. We added this explanation to section 2.2.2, lines 168 to 170. Thanks for bringing this to our attention.

In the revised version of Figure 2 (see comment 9), the added part in the right panel shows how additional boundaries are considered in the hydrodynamic model. We mentioned in the paper that the land boundary is considered greater than the river bank (line 232). One cell was allocated to each defined tributary. This shows that the outlet discharge of the REW, which is close to the

mainstream, was used to define the tributaries in the computational domain of the hydrodynamic model.

Nan, Y., Tian, L., He, Z., et al. (2021). The value of water isotope data on improving process understanding in a glacierized catchment on the Tibetan Plateau. Hydrology and Earth System Sciences. https://doi.org/10.5194/hess-2021-134

Cui, T., Li, Y., Yang, L., et al. (2023). Non-monotonic changes in Asian Water Towers' streamflow at increasing warming levels. Nature Communications, 14(1), 1176.

Sun, Y., Tian, F., Yang, L., et al. (2014). Exploring the spatial variability of contributions from climate variation and change in catchment properties to streamflow decrease in a mesoscale basin by three different methods. Journal of Hydrology, 508, 170-180.

Mou, L., Tian, F., & Hu, H. (2009). Artificial neural network model of runoff prediction in high and cold mountainous regions: A case study in the source drainage area of Urumqi River. Journal of Hydroelectric Engineering, (1), 64-69.

Xu, R., Hu, H., Tian, F., et al. (2019). Projected climate change impacts on future streamflow of the Yarlung Tsangpo-Brahmaputra River. Global and Planetary Change. https://doi.org/10.1016/j.gloplacha.2019.01.012

**6) I was unable to open the website https://portal.mrcmekong.org/home successfully, whether using the network service from German, Japan or China. Perhaps the author could consider uploading the data to such as https://zenodo.org/ for safekeeping.**

**Response:** Thank you for your comment. We have recently been informed that the MRC website was hacked, and MRC technicians are currently working to resolve the issue. This information came to our attention during recent communications with MRC and MRCS (Dr. Sarann Ly), who is a co-author of this paper.

We apologize for the oversight regarding our statement in the "Data Availability" section. While we mentioned that data are publicly available, it has come to our attention that they are accessible upon payment, after which researchers receive a license to use the data for research purposes. Due to this limitation, we are unable to upload the data:

*"In accordance with the MRC Procedures for Data and Information Exchange and Sharing of 01 November 2001, the MRC Secretariat is the Custodian of the MRC Information System. The Licensee has requested and the Licensor – MRCIS Custodian - is prepared to grant a non-exclusive, non-transferable license to the Licensee to use the Licensed Data for the purposes specified in this Noncommercial Data Use License subject to the terms and conditions contained herein".*

**7) At line 88, Firstly, the official name of this basin is the Lancang-Mekong River Basin, with upstream Lancang River and downstream Mekong River. Secondly, the length of the river claimed by the author is questionable. Finally, the number of Chinese reservoirs is more than 11 and needs further verification. Considering that the collaborators include a large number of senior Chinese experts in hydraulic research, it is unacceptable to make mistakes in these details and data.**

**Response:** Thanks for your comment. We agree with the reviewer and accordingly will use "Lancang-*Mekong River Basin*" in the manuscript.

Regarding the length of the river, we acknowledge the varying estimates in the literature. To address this, we will revise the manuscript to state that the length of the river is approximately 4800 km (line 90 of the revised paper)

Regarding the number of large hydropower dams on the mainstream of the Lancang course until 2020, we confirm that there are 11 such dams. We understand that there may be a misunderstanding, and we apologize if the term "mainstream" was not explicitly used in line 92 of the submitted paper. We have revised the sentence to clarify that we are referring specifically to mainstream hydropower dams (see line 95 of the revised paper). However, Figure 1 and its caption indicate the presence of these 11 mainstream dams. Thank you for your feedback.

**8) In Figure 1, what is "the Tonle Sap Lak"? It is recommended that the author carefully checks for the spelling and grammatical errors in the paper, as similar situations occur frequently.**

**Response:** The first author apologizes for the oversight. It should indeed be "Tonle Sap Lake." Below is the updated version of Figure 1.

[Figure]

**9) In Figure 2, I don't think it's a good idea to use both red circles and triangles for labeling. I can't distinguish the tributary station and the mainstream station at all. Besides, I think there should be a space separating the "Delft3D".**

**Response:** Thanks for your feedback and suggestions. We use different colors for labeling as you've suggested. The updated version of Figure 2 is as follows:

Regarding the usage of "Delft3D," we followed the format used on the official website and in the user manual, where it is written as "Delft3D." We have maintained this formatting accordingly in the revised manuscript (see line 224 of the revised paper).

[Figure]

**Figure 2:** Illustration of developed integrated modeling framework. (a) The THREW hydrological model applied to the LMRB. (b) The defined computational domain (white splines, i.e., land boundary) in the developed hydrodynamic model for analyzing daily river flow fluctuations. Each tributary is represented by a single cell located between the land boundary and the riverbank. Note: The cells in panel (b) do not represent the actual number of cells used in the simulations. The name of each defined sub-basin in this study is based on its upstream and downstream stations (panel (a)).

**10) Line 220, "Comparable levels of accuracy are achieved for the years 2019 and 2020, as detailed in the SM, Section 3". My understanding is that you cannot prove the overall model usability by showing only a part. ST is located downstream and has a large main stream flow, making it less affected by reservoir operation. Therefore, using flow velocity assessment at a**

**monthly scale during the rainy season can give better results, but this cannot prove the applicability of the model for basin-wide flow assessment after 2010.**

**Response:** Thank you for your comment. We stated in the manuscript that the basin suffers from insufficient data for tributaries and velocity measurements. Otherwise, we would have been able to provide more comparisons for velocity.

We believe that the ST station can be influenced by its surrounding sub-basins and upstream sub-basins. The significant river flow changes presented in Figure S12g highlight the pronounced contribution of sub-basins to these changes. The ST station is particularly influenced by the 3S basin, which is the most important sub-basin to the Mekong, contributing up to 20% of its flow and providing more runoff than other sub-basins. This sub-basin produces, on average, around 6600 m³/s of runoff to the ST station (see Zhang et al., 2023).

Additionally, achieving such high accuracy in velocity at this station means that the input data (e.g., from the THREW model) for an extensive number of upstream tributaries is of high accuracy; otherwise, it would be challenging to obtain such precision using low-accuracy upstream data. The developed hydrodynamic model has produced time series discharge and water levels at all mainstream stations, including the ST station, with high accuracy (NSE > 0.94).

Given these factors and the uncertainties that not only our study faces but also many other models of the Mekong basin, such as data and DEM inaccuracies, achieving MRE values of less than 7.1% based on point-by-point comparisons in different years (mega-dam period) demonstrates good accuracy and reliable modeling (Figure S4).

Please note that the reason we provided point-by-point comparisons for water level and discharge at other stations like Chiang Khan and Pakse (Figure 4) was to illustrate the accuracy of the developed model in simulating large daily river flow changes that occurred over consecutive days or in a short period. Our analysis is event-based rather than time series-based. Such accurate modeling of events cannot be easily observed in time series graphs like Figure 3.

Zhang, K., Morovati, K., Tian, F., Yu, L., Liu, B., Olivares, M. A. (2023). Regional contributions of climate change and human activities to altered flow of the Lancang-mekong river. Journal of Hydrology: Regional Studies, 50, 101535.

**11) The results in Figure 8 seem to be based on the comparison between the actual observed flow and the natural flow simulated by the model, or did the authors include a simulation of reservoir operation in the model? I am not sure if I missed the part about the reservoir being set up in the model. It is recommended that the authors explain how the results were obtained.**

**Response:** Thank you for your comment. This figure does not depict comparisons between actual observed flow and natural flow simulated by the model. Instead, it presents results obtained through our developed modeling framework. We used measured data for inlet and outlet boundaries and modeled discharge using the THREW model for tributaries, which accounts for 85 dams in the model setup.

In the corresponding section and Figure caption, we specified that these results represent averaged outcomes of all large river flow changes. For instance, at the CS station, there were 22 events recorded from 2010 to 2020, with 58% indicating the average contribution of the sub-basin for these 22 events.

**12) In Figure 9, I don't think it's a good idea to remove the year labels on the x-axis of the time series graph, as this only makes the figure harder to understand. Also, what is "recieved rainfall"? It is recommended to avoid the use of rainfall and to use precipitation uniformly. I suggest that the author consider further detailed checks on the grammar, fonts, font size, etc. of the full text and images. The current version has too many errors.**

**Response:** Thanks for your feedback and suggestions. We have added the years to the x-axis.

"Received rainfall" refers to the cumulative precipitation the corresponding sub-basin receives during the travel time (response time). As highlighted in the paper, this value does not precisely indicate the precipitation received during the travel time but rather provides an approximate representation of the precipitation pattern during that period (see lines 396-397).

We have used "precipitation" instead of "rainfall" as suggested. Regarding the font sizes, we acknowledge that there are discrepancies among figures, and we have made specific adjustments accordingly. Figure 9 has been updated accordingly.

[Figure]

**13) In Figure 10, Same as above, what is "contrinution"? I can understand that the author has a few singular and plural errors or tense problems in the manuscript. However, repeated typing errors in important figures are unacceptable.**

**Response:** The first author apologizes for the oversight. The word "contribution" indeed should be spelled correctly. We have carefully reviewed the manuscript to eliminate any typographical errors.

**14) In Figure 11, Please add dots of corresponding colors on the basis of the lines in the legend, which can make the image more readable.**

Response: Thanks for your suggestion. We have added the dots in the legend as recommended. Thank you.

[Figure]

**15) In Figures 6 and 9, I am not sure how $R^2$ is calculated, what data are used? Please explain in detail.**

**Response:** Thanks for your comment. We did not calculate $R^2$ for Figure 9. In Figure 6, we utilized the observed upstream discharges as inlet boundaries for the hydrodynamic model and calculated the response time for various events. Subsequently, we used the observed discharge and the derived response time to generate correlations in Excel. We determined that the "power correlation" best represents the relationship between upstream discharge and the time required for propagation to the downstream station (line 20 of the submitted manuscript). $R^2$ values were calculated for all events at each station using Excel. The same approach was followed in Figure 11. We have added these details in the SM file, section 10.

**16) Sources of meteorological soil and vegetation DEM data used in modelling must be listed in the main text in a clear and detailed manner. Layered citations are unacceptable.**

**Response:** Thanks for the suggestion. Soil data were obtained using the global soil database provided by the Food and Agriculture Organization of the United Nations (FAO), with a spatial resolution of 10 × 10 km. DEM data were obtained from SRTM (Shuttle Radar Topography Mission), with a spatial resolution of 250 m.

This information has been added to the revised manuscript, section 2.2.1, lines 141 to 145.

---

## Author Response (AR2)

We would like to express our thanks for the positive comments and the valuable questions/suggestions on our manuscript. We have revised the manuscript thoroughly based on all the comments. The reviewer's comments are enumerated. Our replies to each comment start with "Response."

==================================================================

**Editor's Comment**

**The two reviewers appreciate the revisions of your manuscript. They flagged a few more issues that I would request you to address in a further round of minor revisions. I am looking forward to receiving a revised version of your manuscript.**

**Response:** Thank you very much for your feedback and for managing the review process of the paper. Your insights are invaluable, and we appreciate your efforts in this regard.

**Reviewer 1**

**Thanks for the efforts made by the authors, all my concerns were well addressed. I did not have any further comments on this excluding two small issues: (1) No legend in Figure 3; (2) The legend in Figure S4 is too small.**

**Response:** Many thanks for your positive feedback and valuable comments and suggestions during the review process.

We have added a legend for Figure 3 and increased the size of Figure S4 to enhance clarity, as detailed below:

[Figure]

**Figure 3. Comparison of the simulated time series discharge by the THREW hydrological model with measured data over the mega-dam period (2010-2020)**

[Figure]

**Figure S5. A comparison of measured water levels with those obtained by the hydrodynamic model for mainstream stations. CS, CK, NP, MD, PA, and ST represent Chiang Saen, Chiang Khan, Nakhon Phanom, Mukdahan, Pakse, and Stung Treng stations, respectively.**

**Reviewer 2**

**Given that the author has responded in detail to the previous questions, I believe that the current version of the manuscript provisionally meets HESS publication standards. However, before approval for publication, I think the author needs to address a number of issues.**

**Response:** Many thanks for your valuable comments and suggestions during the review process.

**1) In Figure 1, it is recommended to add "Basin" after the word "Lower Mekong" to correspond to "Lancang Basin".**

**Response:** Thank you for your suggestion. We have added the word "basin" to the legend of Figure 1 for clarity, as follows:

[Figure]

**2) Please check the chapter number. After 2.2.2.1 Reservoir Module, I could not find the content of 2.2.2.2.**

**Response:** Thank you for your comment. The description of the hydrological model can be found in Section 2.2.2, with the dam module discussed in Sub-section 2.2.2.1. As such, we do not include Sub-section 2.2.2.2 in the previously submitted revised version.

**3) After the legend and scale of Figure 5 overlap, readability is poor. It is recommended to add a background or to display it separately outside the figure.**

**Response:** Thank you for your comment. We have added a background to the legend to improve readability. The same scale has been used for both figures, as follows:

[Figure]

(a)

(b)

**4) In Figure 7 it is recommended to increase the length of the y-axis. Currently PA-ST is covered by the dark bar. I also noticed that the spacing of the DRY WET labels at the bottom is uneven. Although I am not against using tools such as PS to improve the image, it is advisable to consider the overall aesthetics of the image.**

**Response:** Thank you for bringing this issue to our attention. We have adjusted the spacing between labels and increased the length of the y-axis to enhance the visualization of labels and percentages for the PA-ST sub-basin. Thank you for your feedback.

[Figure]

**5) In Figure 8 it is recommended to mark the numbers at the top of the BAR and display them in the centre. The current display effect may mislead people into thinking that the digital labels only represent the impact of the dam, without considering the impact of the upstream reservoir.**

**Response:** Thank you for your suggestion. We have moved all numbers to the top and center of the bars for improved clarity. As we highlighted in various sections of the paper and in the caption for Figure 8, the percentages represent the contribution of each sub-basin to its downstream station. We did not separate the impacts of dams and precipitation for each sub-basin (see lines 449 to 454). Additionally, as mentioned in Section 3.1.2, our objective is to conduct a regional contribution analysis. In Section 2.3, we provide definitions for each period, which refer to human activities such as dam construction, land cover changes, and irrigation projects.

[Figure]

**6) For Figure 10, I suggest adding a black horizontal line at the 0 scale, and the 0 values of the Y axis on both sides should be aligned. In addition, I am not sure that the rightmost column of PA is displayed correctly.**

[Figure]

**7) I suggest the author check the grammar and tense of their text before final acceptance before publication.**

**Response:** Thank you for your suggestion. We have carefully reviewed the grammar and tense of the text, making necessary adjustments to enhance clarity. Please refer to the revised manuscript for the changes. Thank you.

---

## Author Response (AR3)

We would like to express our thanks for the positive comments and the valuable questions/suggestions on our manuscript. We have revised the manuscript thoroughly based on all the comments. The reviewer's comments are enumerated. Our replies to each comment start with "Response."

==================================================================

**Editor's Comment**

**Thank you very much for the additional revisions.**

**In principle, I will be glad to accept the manuscript for publication in HESS. However, please incorporate the last necessary technical corrections associated to the maps as indicated in the message by the Editorial team.**

**Response:** Thank you very much for your feedback and for managing the review process. Your thoughtful consideration of the paper is greatly appreciated, and your insights are invaluable to us.

**The comment raised by the Editorial team:**

**I just noticed that your figure 2b contains a map. To clarify whether a copyright statement or a credit must be given in the map itself or in the caption, we differentiate between (a) maps entirely created by you, (b) maps created by you but based on layers reused from other originators, or (c) maps simply reused from other originators. An example for (a) is a digital elevation model (DEM) purely based on measurement points collected by you and derived by using a software product. If you use an existing map layer from another originator as a basis for significantly enriching the map with your own content, this would be an example for case (b). Case (c) could be a pure reproduction of Google Maps where your own contribution is rather small (e.g. a city map where you only added a few marks for your study locations). If the map was entirely created by you (case a), there is no need to change the caption or map. Please simply inform us. To the contrary, if your map follows cases (b) or (c), please let us know whether the map is distributed under public domain. If yes, please do not include a copyright statement (copyright is waived) but consider adding a credit to the map or caption. However, if your map follows cases (b) or (c) and is not distributed under public domain, please include at least a credit or even a copyright statement (e.g. © Google Maps), if this is required by the map provider, in the map itself or in the caption.**

**Response:** Thanks for your comment. Regarding your request for a copyright statement for Figure 2b of the manuscript numbered "hess-2024-96," I found that this figure could be classified as either case (b) or (c). After reviewing the website, I determined that we do not need to obtain permission. However, it would be prudent to include a statement in the caption, as you suggested. We have added the following statement, which reads as follows.

**Figure 2. Illustration of developed integrated modeling framework: (a) the THREW hydrological model applied to the LMR basin: (b) the defined computational domain (white splines, i.e., land boundary) in the developed hydrodynamic model for analyzing daily river flow fluctuations. Each tributary is represented by a single cell located between the land boundary and the riverbank (red cells in panel (b)). Note: the cells in panel (b) do not represent the actual number of cells used in the simulations (see Section 4 of the SM file for more details). The name of each defined sub-basin in this study is based on its upstream and downstream stations (panel (a)). The background map in Figure 2b is adapted from Google Earth Maps (e.g., © Google Maps).**